# Small Transformers Don't Need LayerNorm at Inference Time: Scaling LayerNorm Removal to GPT-2 XL and Implications for Mechanistic Interpretability

**Luca Baroni** *
Charles University
baroni@ksvi.mff.cuni.cz

**Galvin Khara**\*
Imperial College London
gk2510@ic.ac.uk

**Joachim Schaeffer**\*
TU Darmstadt / MIT
joachim.schaeffer@gmail.com

**Marat Subkhankulov**\*
Independent
m.subkhankulov@gmail.com

**Stefan Heimersheim**
Apollo Research
stefan@apolloresearch.ai

## Abstract

Layer-wise normalization (LN) is an essential component of virtually all transformer-based large language models. While its effects on training stability are well documented, its role at inference time is poorly understood. Additionally, LN layers hinder mechanistic interpretability by introducing additional nonlinearities and increasing the interconnectedness of individual model components. Here, we show that all LN layers can be removed via fine-tuning from every GPT-2 model with only a small increase in validation loss (e.g. +0.03 cross-entropy loss for GPT-2 XL). Thus, LN is not essential at inference to maintain comparable performance in these models. We find that the amount of fine-tuning data needed for LN removal grows sublinearly with model parameters, suggesting scaling to larger models is feasible. We release a suite of LN-free GPT-2 models on Hugging Face. Furthermore, we test interpretability techniques on LN-free models. Direct logit attribution now gives the exact direct effect of individual components, while the accuracy of attribution patching does not significantly improve. We also confirm that GPT-2's "confidence neurons" are inactive in the LN-free models. Our work clarifies the role of LN layers in language modeling, showing that GPT-2-class models can function without LN layers. We hope that our LN-free analogs of the GPT-2 family of models will enable more precise interpretability research and improve our understanding of language models.[1]

## 1 Introduction

Large language models (LLMs) have seen widespread adoption in recent years (Touvron et al., 2023; OpenAI et al., 2024; Gemini Team et al., 2024), most of which are based on the Transformer architecture (Vaswani et al., 2017). A key component of virtually all such LLMs are layer-wise normalization (LN) layers, often implemented via LayerNorm (Ba et al., 2016)

$$\text{LN}(\mathbf{x}) = \frac{\mathbf{x} - \mu}{\sigma} \odot \boldsymbol{\gamma} + \boldsymbol{\beta}, \quad \mu = \frac{1}{H}\sum_{h=1}^{H} x_h, \quad \sigma = \sqrt{\frac{1}{H}\sum_{h=1}^{H}(x_h - \mu)^2 + \epsilon}, \tag{1}$$

where $\odot$ denotes the Hadamard (element-wise) product. A common alternative is RMSNorm (Zhang & Sennrich, 2019), which performs an equivalent computation, without removing the component in

---

*These authors contributed equally

[1]A precursor of this work has been presented at the *Interpretable AI: Past, Present and Future* workshop at NeurIPS 2024, under the title "You can remove GPT2's LayerNorm by fine-tuning" (Heimersheim, 2024)

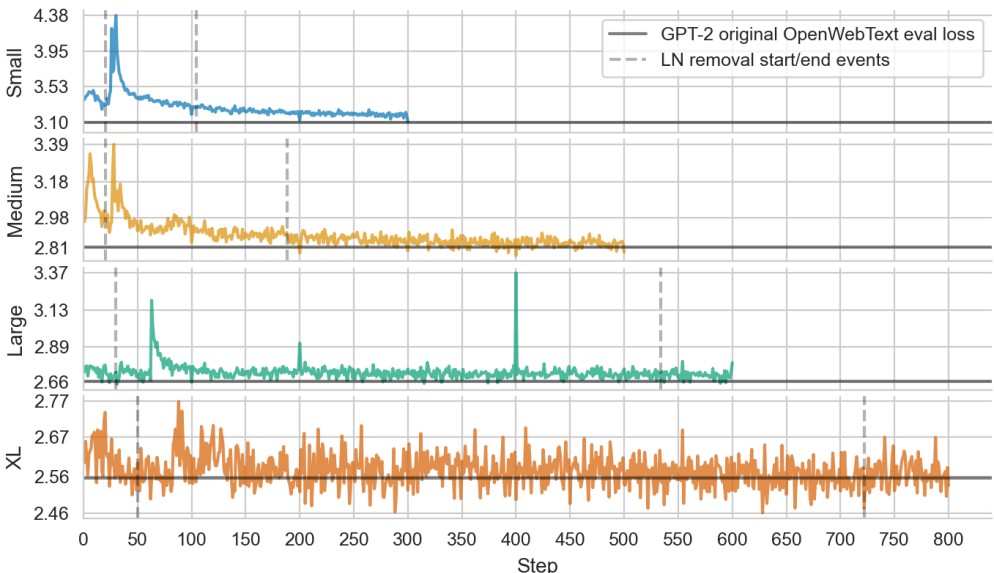

Figure 1: Main training loss curves for all GPT-2 variants during LN removal. Original GPT-2 OpenWebText eval losses are shown for reference. Curves terminate at model suite checkpoints. LN removal period shown as vertical lines.

the $(1, 1, \ldots, 1)$ direction (see Appendix D for a more in depth comparison). These layers have been introduced to stabilize the training process (Ba et al., 2016), similar to batch normalization (Ioffe & Szegedy, 2015) in other network architectures.

Unlike batch normalization however, LN layers cannot be replaced with a linear transformation at inference time. While the mean centering ($\mu$), weight ($\gamma$), and bias ($\beta$) parameters can be folded into neighboring layers (e.g. `fold_ln`, Nanda & Bloom, 2022), the non-linear division by the norm or standard deviation of the residual stream must be executed at inference time. This raises the question of what role LNs play in the model and whether it is necessary for the model to function. Prior work has shown that LN functions can implement complex non-linear functions in toy models (Winsor, 2022), and proposed that LNs might play a role in confidence regulation in LLMs (Stolfo et al., 2024).

Additionally, LN layers complicate mechanistic interpretability. Mechanistic interpretability typically aims to decompose the model into smaller components and to understand their individual effects and interactions. Both of these are complicated by the non-linearity of LN layers. Individual components cannot be easily attributed as their effect on LN depends on the residual stream activations (nostalgebraist, 2020; Elhage et al., 2021; Wang et al., 2022b; Nanda, 2023b;a). Interactions between components are also complicated by LN because it causes each component to affect almost every downstream component in the model (via the LN scale). This makes analyzing the interactions complex (e.g. Bushnaq et al., 2024; Farnik et al., 2025). In practice, researchers approximate the LN layers as linear transformations (referred to as "freezing LayerNorm"; Bricken et al., 2023; McDougall et al., 2023; Kissane et al., 2024), or train models without LN layers (Elhage et al., 2021; Nabeshima, 2024).

In this work we show that LN layers can be removed from transformer models at the end of training. We replace the LN layers with a linear transformation that is initialized to be close to the original LN transformation, and fine-tune the model on a small fraction of its training data. We do this for one LN layer at a time, essentially slowly weaning the model off of LN. This (a) shows that LLMs can function without LN layers, and (b) provides a LN-free versions of the GPT-2 family of models. These models can be studied on their own, simply to understand any large language model, or as a proxy for their corresponding original GPT-2 models. The latter is possible as our fine-tuned models have similar internals, but should be used with caution as similarity is not exact.

Our contribution is threefold:

- We show that LLMs can function without LN layers, achieving a cross-entropy loss comparable to the original models.

- We provide an optimized protocol for removing LN layers from LLMs at the end of training or during fine-tuning, and provide a suite of LN-free GPT-2 models on Hugging Face.

- We validate that the interpretability of LN-free models is improved, finding that the direct logit attribution (DLA) error is reduced from 50% to 0%, and that attribution patching—contrary to expectations in the literature—does not improve with LN-free models.

## 2 RELATED WORK

**Mechanistic interpretability:** Interpretability aims to understand the internals of neural networks and the algorithms they implement. Most mechanistic interpretability methods attempt to decompose a model into smaller components and aim to understand the interactions between those components. The most popular methods are based on sparse dictionary learning, such as sparse autoencoders (Bricken et al., 2023) or cross-layer transcoders (Ameisen et al., 2025). In both cases, researchers attempt to find a sparsely-interacting set of components that explain the model's behavior (Marks et al., 2024; Lindsey et al., 2025). The most common approach to deal with LN is to approximate the layer norm scale as constant (e.g. Bricken et al., 2023; McDougall et al., 2023; Kissane et al., 2024). Other methods introduce special cases for LN layers (e.g. Bushnaq et al., 2024).

**LN alternatives:** The main alternative to layer normalization is batch normalization (BN). However, BN performs worse than LN in language model transformers due to changes between the training and inference distributions (e.g. Wang et al., 2022a).

Concurrent work (Zhu et al., 2025) proposed a Dynamic Tanh (DyT) as an alternative to normalization. Instead of an LN layer, they apply an element-wise $\tanh(\alpha x)$ function to the residual stream. This work confirms our results, finding that language models can work without LN. While DyT is preferable over LN, in some use cases, DyT is still a non-linear function whose role we don't understand, and that affects interpretability. Our work goes further, replacing LN with a purely linear transformation.

**Transformers trained without normalization:** Finally, Nabeshima (2024) trains toy language models from scratch, without normalization. However, we expect this method to work only for small language models, state-of-the-art language models continue being trained with normalization. Thus we focus on removing LN from an already-trained model.

## 3 LN REMOVAL STRATEGY AND METHODS

We remove the nonlinearity of LN by replacing the standard deviation in (1) by a scalar, corresponding to an estimate of the average standard deviation, $\overline{\sigma}_{\mathrm{avg}}$, while fine-tuning on OpenWebText. We define a FakeLN block as

$$\text{FakeLN}(\mathbf{x}) = \frac{\mathbf{x} - \mu}{\overline{\sigma}_{\mathrm{avg}}} \odot \boldsymbol{\gamma} + \boldsymbol{\beta}, \quad \sigma_{b,s} = \sqrt{\frac{1}{H}\sum_{h=1}^{H}(x_{b,s,h} - \mu_{b,s})^2 + \epsilon}, \quad \sigma_{\mathrm{avg}} = \frac{1}{BS}\sum_{b=1}^{B}\sum_{s=1}^{S}\sigma_{b,s},$$

(2)

where $\sigma_{b,s}$ is the standard deviation across the model dimension for batch index $b$ and sequence position $s$, and $\sigma_{\mathrm{avg}}$ is the average across all tokens in a batch. $\overline{\sigma}_{\mathrm{avg}}$ is the fixed scalar value used when replacing LN with FakeLN. Because removing all LN blocks simultaneously irreparably breaks the model's performance, we adopt a sequential removal process during fine-tuning: we remove one LN block, fine-tune for a fixed number of steps to stabilize the loss (which typically spikes after each removal), and then proceed to the next LN block. Furthermore, $\sigma_{\mathrm{avg}}$ can drift during fine-tuning. Therefore, to minimize the disruption introduced by LN removal and stabilize the fine-tuning process, we recompute $\sigma_{\mathrm{avg}}$ for each batch and freeze the scale factor in FakeLN at the moment of removal to $\overline{\sigma}_{\mathrm{avg}} = \sigma_{\mathrm{avg}}$. For the small and medium models, the batch size is significantly large enough to produce reliable estimates of $\sigma_{\mathrm{avg}}$. For GPT-2 Large and GPT-2 XL, we use an exponential moving average filter to update $\sigma_{\mathrm{avg}}$ for new batches. After LN removal, $\overline{\sigma}_{\mathrm{avg}}$ is not updated anymore.

We categorize LN blocks into $\text{LN}_{\text{qk}}^l$, $\text{LN}_{\text{v}}^l$, $\text{LN}_{\text{MLP}}^l$ and $\text{LN}^f$, where $l$ indicates the layer number. Respectively, these LN blocks normalize inputs to the query/key path, the value path, the MLP, and the final unembedding. While splitting LN for attention heads paths is uncommon, we find this more fine-grained removal of LN improves stability during fine-tuning. Our sequential removal process begins after an initial standard fine-tuning phase with the removal of $\text{LN}_{\text{MLP}}^0$, followed by $g_{\text{mlp}}$ fine-tuning steps. We then remove $\text{LN}_{\text{MLP}}^1$, fine-tune again for $g_{\text{mlp}}$ steps and continue this pattern layer by layer until $\text{LN}_{\text{MLP}}^L$ is removed where $L = N_{\text{layers}}$. We then apply the same pattern to remove $\text{LN}_{\text{qk}}^l$ and $\text{LN}_{\text{v}}^l$ blocks, each separated by $g_{\text{qk}}$ and $g_{\text{v}}$ fine-tuning steps, respectively. Finally, we remove $\text{LN}^f$. The gaps between removal events are hyperparameters that have to be chosen carefully. Too small gaps can result in instabilities, while choosing very large gaps results in unnecessarily high computational costs. We provide a table with LN removal schedule and more details in Appendix B.

We empirically found that beginning with $\text{LN}_l^{\text{MLP}}$ rather than $\text{LN}_l^{\text{qk}}$ led to more stable fine-tuning, likely because residual norm variance at beginning-of-sequence tokens affects the attention mechanism more strongly when its normalization is removed first. Despite removing LN blocks sequentially, instabilities can still occur during LN removal. To further stabilize LN-removal by fine-tuning, we used an additional auxiliary loss.

**Auxiliary Loss**   In models with LN, residual stream vectors are scaled by their standard deviation[2]. When LN is removed, large norm disparities across positions can lead to gradient spikes and destabilize fine-tuning. To encourage stable activations during this process, we introduce an auxiliary loss that promotes consistent standard deviations across token positions:

$$\mathcal{L}_{\text{aux}} = \lambda \cdot \mathbb{E}_{b,s}\left[(\sigma_{b,s} - \hat{\sigma})^2\right], \qquad \hat{\sigma} = \frac{1}{|\mathcal{M}|} \sum_{(b,s)\in\mathcal{M}} \sigma_{b,s}, \qquad (3)$$

where $\lambda$ is a scalar hyperparameter. While the loss itself is computed across all positions in the batch, the target $\hat{\sigma}$ is the average standard deviation across the subset of token positions $\mathcal{M}$, excluding the first token (position 0) and any positions containing the end-of-text token (ID 50256). These exclusions from the target calculation are motivated by the observation that such positions consistently exhibit higher variance in GPT-2 models. We apply the auxiliary loss only at $\text{LN}^f$ since all residual streams propagate through this final normalization layer, making it a natural global target for norm regularization.

## 4   LN REMOVAL RESULTS

We successfully remove LN during fine-tuning on OpenWebText from GPT-2 Small, Medium, Large, and XL (Tab. 1), demonstrating that our sequential LN removal strategy with auxiliary loss scales from a 124 million parameter model to a 1.5 billion parameter model. Figure 1 shows the main loss during fine-tuning for LN-removal. Notably, the amount of fine-tuning data required grows sublinearly with model size. For details of the sequential LN-removal schedule, hyperparameters, and fine-tuning data requirements, see Appendix B. We find that the largest main loss spikes appear during the removal of $\text{LN}_{\text{MLP}}$ blocks, which is the first LN block that is removed. The $\text{LN}_{\text{qk}}$ and $\text{LN}_{\text{v}}$ block removals result only in small main loss spikes. Before introducing the auxiliary loss, the LN-removal fine-tuning loss curves were more spiky, suggesting that the auxiliary loss effectively absorbs some of the effects of LN removal. Furthermore, the auxiliary loss decreases quickly at the beginning of fine-tuning, indicating that the model successfully learns to maintain consistent standard deviations across token positions.

As a control, we compare the LN-free GPT-2 model suite to the original GPT-2 models and vanilla fine-tuned models. The vanilla fine-tuned models were fine-tuned for the same number of steps and with the same learning rate schedule as the LN-free models, but without auxiliary loss and without removing LN. This control allows us to disentangle the effects of LN from the effects of fine-tuning.

We evaluate performance using mean cross-entropy loss on a validation set of OpenWebText, The Pile, and The Pile-filtered (Tab. 1). The Pile-filtered consists of sequences from The Pile dataset (monology-pile-uncopyrighted), filtered by removing sequences containing tokens that appear in The

---

[2]After mean-centering, i.e., removing the component in the $(1, 1, \dots, 1)$ direction (Gupta et al., 2025).

Pile but not in OpenWebText, such as control characters which arise from formatting discrepancies between the two datasets (see Appendix E for more details).

We find that LN-free models perform comparably to their original variants, with performance gaps ranging from +0.03 to 0.1 cross-entropy loss difference on The Pile-filtered (Tab. 1). This comparable performance extends to standard language understanding benchmarks, where LN-free models maintain accuracy within 1-2 percentage points from their original variants (Appendix F). The only notable exception is GPT-2 XL LN-free, which shows degraded performance on The Pile. A closer examination of the distribution of losses reveals that the higher averaged CE loss is driven by a very small number of samples and that the 99.9 percentile ranges of GPT-2 XL LN-free and GPT-2 original are nearly identical for The Pile, indicating that the vast majority of sequences are handled similarly by both models. This suggests that GPT-2 XL LN-free is highly overconfident on a small number of sequences that are present in The Pile but absent from The Pile-filtered dataset.

We also investigate whether the performance gap can be closed by simply fine-tuning LN-free models for longer. Contrary to our initial expectations, we find that extending fine-tuning does not reduce the loss gap to vanilla models. Instead, the gap remains approximately constant throughout fine-tuning, suggesting that LN contributes a small but persistent performance benefit that cannot be compensated by additional fine-tuning. We discuss potential mechanisms behind this behavior in Section 5.4. We also investigate if our methodology generalizes beyond the GPT2 model family, and successfully apply our removal strategy to Pythia 70M (Biderman et al., 2023) (see Appendix C).

Table 1: Overview of LN-free, vanilla fine-tuned, and original GPT-2 models. Reported values are mean cross-entropy over 10.2M tokens from The Pile and The Pile-filtered, and 4.5M tokens from the OpenWebText (OWT) validation set. For each model size and dataset, the lowest loss is highlighted in bold, and the loss difference between the LN-free model and the best-performing model is shown in brackets. See Appendix A for Hugging Face links to the models and a summary of compute requirements.

| Model | FT steps | OWT (val) | The Pile | The Pile-filtered |
|---|---|---|---|---|
| GPT-2 Small original | 0 | 3.1006 | **2.8450** | **2.7899** |
| GPT-2 Small vanilla | 300 | **3.0126** | 2.8511 | 2.8112 |
| GPT-2 Small LN-free | 300 | 3.0797 [+0.0671] | 2.8852 [+0.0402] | 2.8757 [+0.0858] |
| GPT-2 Medium original | 0 | 2.8145 | **2.5163** | **2.5390** |
| GPT-2 Medium vanilla | 500 | **2.7390** | 2.5752 | 2.5724 |
| GPT-2 Medium LN-free | 500 | 2.7642 [+0.0252] | 2.6579 [+0.1416] | 2.6352 [+0.0962] |
| GPT-2 Large original | 0 | 2.6623 | **2.5320** | **2.4347** |
| GPT-2 Large vanilla | 600 | **2.6240** | 2.6233 | 2.5074 |
| GPT-2 Large LN-free | 600 | 2.6384 [+0.0144] | 2.7504 [+0.2184] | 2.5159 [+0.0812] |
| GPT-2 XL original | 0 | 2.5567 | **2.4436** [3] | **2.3739** |
| GPT-2 XL Vanilla | 800 | **2.4799** | 2.4673 | 2.3821 |
| GPT-2 XL LN-free | 800 | 2.5052 [+0.0253] | 130.2197 [4] | 2.3992 [+0.0253] |

# 5 MECHANISTIC INTERPRETABILITY ANALYSES ON LN-FREE MODELS

Removing LN eliminates nonlinear dependencies between components and results in models where residual stream directions map linearly to output logits. In this section, we evaluate common interpretability methods, such as Direct Logit Attribution (DLA) (nostalgebraist, 2020; Elhage et al., 2021; Wang et al., 2022b; Nanda, 2023b) and attribution patching (Nanda, 2023a) on LN-free models and compare the results to their counterparts with LN.

---

[3] GPT-2 XL original: Median: 1.0103, 95th perc: [0.0005, 10.6193], 99.9th perc: [≈0.0000, 43.0064]
[4] GPT-2 XL LN-free: Median: 1.0937, 95th perc: [0.0004, 10.7548], 99.9th perc: [≈0.0000, 48.6459]

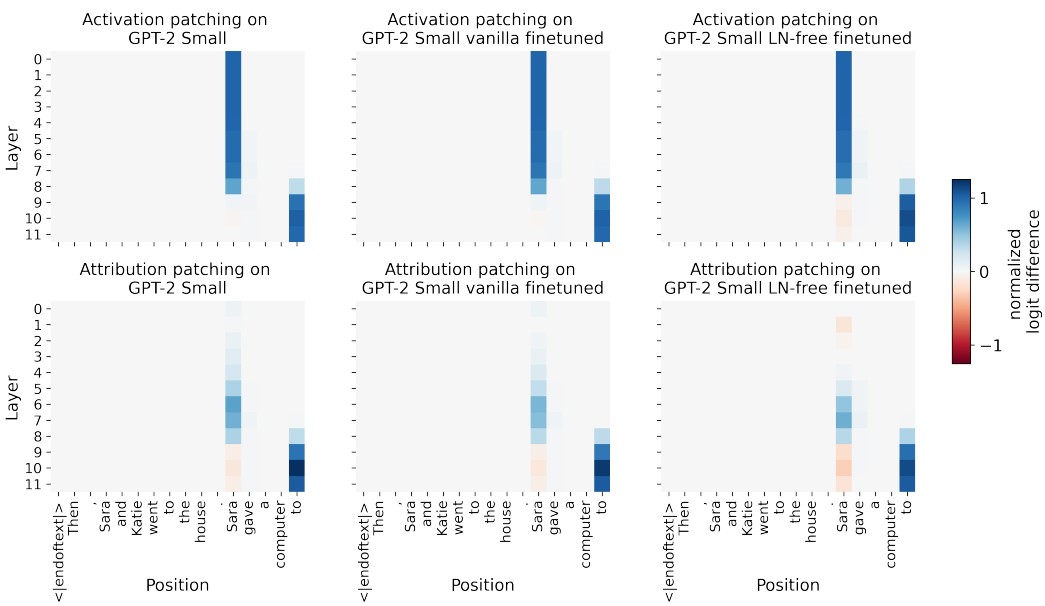

Figure 2: Activation patching and attribution patching applied on the residual stream at different layers and positions on GPT-2 Small and the corresponding vanilla and LN-free versions.

## 5.1 DIRECT LOGIT ATTRIBUTION ON LN-FREE MODELS GIVES EXACT DIRECT EFFECT ON LOGITS

Direct Logit Attribution (DLA) is an approximation to the Direct Effect (DE) of a component. The DE (Pearl, 2022; Geiger et al., 2024) is the effect of a model component on the outputs that is not mediated by intermediate components, and can be computed by subtracting a component's output $c$ from the residual stream $r$ after the final layer, and taking the difference in outputs,

$$\text{DE}(c) = \text{LN}(r) \cdot W_U - \text{LN}(r - c) \cdot W_U, \tag{4}$$

where $W_U$ denotes the unembedding, and LN the final LayerNorm. The DLA approximation is computed using the cached LN scale,

$$\text{DLA}(c) = \text{LN}_{\text{cached}}(c) \cdot W_U, \tag{5}$$

which effectively linearizes LayerNorm (LN).

We calculated both DLA and DE on each attention head in GPT-2 Small original, GPT-2 Small vanilla FT, and GPT-2 Small LN-free FT, on 1,000 sequences consisting of 512 tokens from The Pile-filtered, for logits corresponding to the correct target token. To compare metrics, we used the Normalized Mean Absolute Error (NMAE)[5], which measures the average discrepancy between DLA and DE, expressed as a percentage of the average magnitude of the DE. Our LN-free fine-tuned model achieves a perfect 0.00% [0.00%, 0.00%] (95% Confidence Interval - CI) NMAE, whereas the original model exhibits an NMAE of 49.07% [29.92%, 66.10%]. This result shows that removing LNs makes these methods mathematically equivalent, eliminating the need for inaccurate linearization approximations. See Appendix G for more details.

## 5.2 ACCURACY OF ATTRIBUTION PATCHING ON LN-FREE MODELS DOES NOT SIGNIFICANTLY IMPROVE

Activation patching (Meng et al., 2022; Zhang & Nanda, 2023; Heimersheim & Nanda, 2024) is an interpretability method used to assess the causal roles of neural network components by transferring

---

[5]We calculate NMAE, using averages of absolute differences and DE magnitude rather than per-sample ratios, as we did not observe a consistent proportional relationship between these two measures across samples.

activations from a "clean" prompt that elicits correct model behavior into a "corrupted" prompt that typically leads to incorrect behavior. Formally, this can be expressed as:

$$\Delta = f(x_{\text{corr}}; a_l \leftarrow a_l(x_{\text{clean}})) - f(x_{\text{corr}}), \tag{6}$$

where $f(x)$ measures differences in model predictions (typically logit differences), and $a_l \leftarrow a_l(x_{\text{clean}})$ indicates replacing the corrupted activation with its clean counterpart at layer $l$. While precise, activation patching is computationally expensive, scaling with the number of components tested. Attribution patching Nanda (2023a) addresses this approximating activation patching with a first-order Taylor expansion around the corrupted activation, requiring only two forward passes and one backward pass,

$$\Delta = f(x_{\text{corr}}; a_l \leftarrow a_l(x_{\text{clean}})) - f(x_{\text{corr}}) \approx \nabla_{a_l} f(x_{\text{corr}}) \cdot (a_l(x_{\text{clean}}) - a_l(x_{\text{corr}})) = \Delta_{\text{attr}}. \tag{7}$$

As LN projects residual vectors onto a $(d_{\text{model}} - 1)$-dimensional sphere after removing the mean component, it causes derivatives to vanish when patched directions align with the residual stream and is, therefore, a source of attribution patching errors, i.e. discrepancies between attribution patching estimates and ground-truth activation patching results (Nanda, 2023a, described it for this reason as "*a particularly thorny nonlinearity*").

We investigate whether LN is the primary factor limiting attribution patching accuracy by comparing attribution patching across three models: GPT-2 Small, the corresponding LN-free fine-tuned, and vanilla fine-tuned. We focused on the residual stream preceding each transformer block, a location where attribution patching is known to perform particularly poorly in models with LN. We used 480 IOI Wang et al. (2022b) prompts, systematically varying names, places, and objects, with each prompt paired with counterparts covering all possible name orderings. To ensure alignment across inputs, all prompts had fixed token lengths and name positions. We applied both techniques and quantified how well attribution patching approximates activation patching across layers. We used normalized logit differences as the patching metric to enable robust comparisons across methods. Surprisingly, attribution patching yielded very similar results across layers in the three models (see Fig. 2) and despite removing LN, we observed no improvement in attribution patching accuracy. For each layer, we quantified this by computing the sum of absolute attribution patching errors across token positions in the vanilla fine-tuned model, and subtracting the corresponding value from the LN-free model. This yielded a per-layer improvement score, where positive values indicate lower attribution error in the LN-free model. Averaged across layers, the improvement is $\mu = -0.026$, with standard deviation $\sigma = 0.082$. This negative but informative result suggests that attribution patching's limitations likely arise from other more fundamental nonlinearities in the transformer architecture, namely the attention SoftMax or the MLP activation functions.

### 5.3 FIRST POSITION TOKENS ARE NO LONGER SPECIAL

A well-documented phenomenon in transformer-based language models is the disproportionately high L2 norm of first position token's hidden representations (Xiao et al., 2024; Yona et al., 2025; Barbero et al., 2025). This characteristic has been identified as a key mechanism behind "attention sinks," where the first token captures an outsized portion of attention across multiple heads, affecting information flow throughout the network. While this mechanism appears to help standard models avoid representational collapse by controlling information mixing across layers, it introduces computational irregularities and potential vulnerabilities (Yona et al., 2025).

To investigate whether our models exhibit similar behaviours, we measured the L2 norm of first position tokens, compared to all other tokens, on 1,000 sequences consisting of up to 512 tokens from The Pile-filtered. LN-free models reveal a disruption of the typical first position token norm pattern. As illustrated in Fig. 3, the LN-free model maintains consistently moderate L2 norm values ($\sim$300 to 500) across all layers for the first token, in contrast to the significant norm inflation observed in models with LN. This more uniform norm across token positions represents a fundamental shift from the standard architecture, where the first token's norm typically exceeds that of other tokens by close to an order of magnitude. The largest first token norm growth in all three models was due to the attention head in layer 3, where norms grow from $\sim$500 to 3,600 for the models with LN.

We also investigated the attention sink rate across models, defined as the proportion of attention heads where the first token attracts at least 30% of overall attention. For the original model, the sink rate was 55.3% [53.1%, 58.1%] (95% CI), which dropped to 45.3% [42.0%, 48.5%] for our LN-free

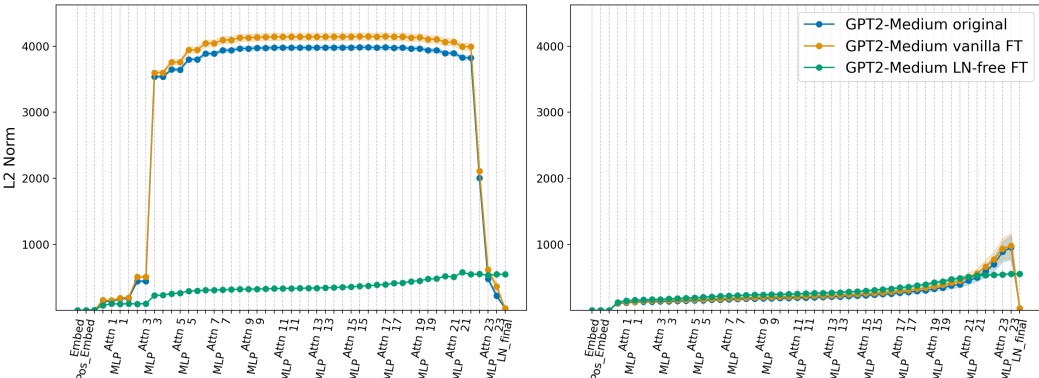

Figure 3: L2 norm growth for first position tokens (left) versus other positions tokens (right) for GPT-2 Medium models. First token norms significant deviate from norms at other positions for models trained with LN. LN-free model treats first token norm similarly to other positions.

variant. Interestingly, while this represents a notable reduction in sink rate, it is not proportional to the reduction we observed in L2 norms. This suggests that the relationship between relative token norm magnitudes and attention sink behavior is likely complex, with attention mechanisms potentially maintaining some degree of positional bias toward the first token even when its norm is substantially reduced.

This effect is likely due to the constant linear scaling applied by FakeLN. In models with LN, residual stream vectors are scaled by their individual standard deviations, meaning components are trained to operate under normalized input conditions. Once LN is removed, this normalization is no longer enforced. To compensate, the model appears to adapt by reducing variability in token norms, such as between the first token and the rest of the sequence. Our auxiliary loss further encourages norm consistency by explicitly penalizing variation across positions, however, we did observe this fundamental change in norm behavior even in experiments without this loss term.

## 5.4 CONFIDENCE NEURONS ARE NEUTERED IN LN-FREE MODELS

When developing our LN-free model variants, we observed a consistent pattern: models exhibited significant overconfidence compared to their original counterparts. For GPT-2 Medium, the average entropy of the output distribution decreased from 2.86 [2.86, 2.87] (95% CI) in the original model to 2.53 [2.52, 2.54] in the LN-free version. Correspondingly, the expected calibration error, defined as the average absolute difference between the predicted confidence and accuracy, increased from 0.019 [0.018, 0.020] to 0.034 [0.033, 0.035]. Motivated by these observations, we investigated how the recently discovered "confidence neurons" (also referred to as "entropy neurons") (Katz & Belinkov, 2023; Gurnee et al., 2024; Stolfo et al., 2024) in the final MLP layer were affected by our LN removal strategy.

Following Stolfo et al. (2024), we define confidence neurons as neurons in the final MLP with (a) a high weight norm, and (b) a uniform impact on all output logits. We detail how confidence neurons were identified and further analysis in Appendix H. We identified the same top-3 confidence neurons (1083, 1108, 3144) in GPT-2 Medium original, vanilla FT, and LN-free. To measure their importance in each model, we conducted mean ablations on 1,000 sequences consisting of 512 tokens in The Pile-filtered. For each neuron $i$, we replaced its input activation with its mean value across the dataset ($x_i \rightarrow \mathbb{E}[x_i]$). This intervention removes the neuron's contextual information while maintaining its average contribution. Figure 4 highlights the absolute change in cross-entropy loss when mean ablating each neuron. In the GPT-2 Medium original, all three neurons increase CE loss when ablated, with neuron 3144 showing the largest effect. In contrast, the impact is completely eliminated in LN-free model. This confirms that linearizing LN completely disables entropy neurons in the final MLP layer, further supporting previous work that identified LN's non-linearity as their primary enabling mechanism (Stolfo et al., 2024). We also observed a decrease in the effectiveness

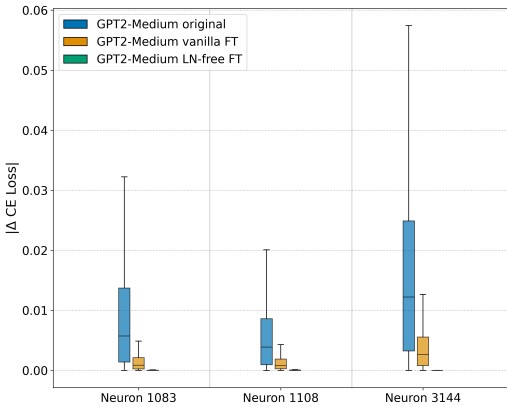

Figure 4: Absolute change in cross-entropy (CE) when ablating top-3 confidence neurons in GPT-2 Medium models. GPT-2 Medium original demonstrates a significant change in CE loss upon ablating, effect is significantly dampened in vanilla FT, and completely disappears in LN-free.

of confidence neurons in our vanilla FT model, likely due to our fine-tuning hyperparameters, which we discuss further in Appendix H.

# 6 DISCUSSION

## 6.1 LIMITATIONS

We successfully remove LN from all GPT-2 models. Here, we want to highlight common issues and possible limitations of this process. We find that the fine-tuning process when LNs are partially removed is, as expected, less stable. We find that the training loss can spike to high values on some inputs, which sometimes causes the training run to fail (irrecoverably high loss). A common failure we observed are exploding gradients, which most often occur during $LN_v^l$ removal. Instabilities usually appear as a cascade of increasing gradient norms or exploding gradients in a single step.

While our LN-removal strategies developed on GPT-2 Small and Medium largely transfer to the Large and XL models, they required significant hyperparameter tuning. Additionally, an early version of our protocol without auxiliary loss worked for GPT-2 Small, but did not scale to larger models, suggesting that protocols don't always generalize across models.

As highlighted in Section 5.4, all of our LN-free models exhibit overconfidence compared to their LN counterparts. While our experiments demonstrate that removing LN effectively neutralizes confidence neurons, the magnitude of the observed increase in overconfidence suggests additional contributing factors. It's possible that without the normalizing effect of standard LN, attention, and MLP components must now handle greater variability in residual stream inputs, potentially compromising their ability to contribute to appropriate uncertainty quantification.

Finally, we note that the LN-free models we open-source are not easily quantizable. We do not consider this a significant limitation as quantization is uncommon in mechanistic interpretability research, the primary intended use case for these models.

## 6.2 FUTURE WORK

**More models:** We focused primarily on GPT-2 models, due to their ubiquity in the interpretability community. In the future, we would like to expand our LN removal protocol to more recent models (see Appendix C and D on our expectations regarding generalizations to different architectures).

**Parameter efficient fine-tuning:** So far we used full fine-tuning. While this was feasible for GPT-2 sized models, we want to explore parameter efficient fine-tuning strategies in the future.

**Further protocol optimization:** We noticed that the gap between removing the LN in different layers can be reduced for $LN^l_{qk}$ and $LN^l_{MLP}$; in fact some experimental runs showed that we could remove those instances of LN in all layers simultaneously (only $LN^l_v$ always required gaps).

**Circuits interpretability:** Attempts to create a sparse computational graph to represent a neural network are hindered by LN. It would be interesting to see if techniques like Marks et al. (2024) benefit from removing LN layers.

## 7 CONCLUSIONS

We showed that layer normalization can be gradually removed from transformer models with minimal performance loss using a fine-tuning procedure, demonstrating this on all GPT-2 models (and Pythia-70M). Our results demonstrate that nonlinearities such as LN are not necessary for GPT-2-class transformers to effectively model language at inference time. We detailed our procedure and the strategies used to address hyperparameter sensitivity. Applying interpretability techniques, we found that in LN-free models DLA becomes an exact estimate of DE and first-token residual norms become comparable to those at other positions. Surprisingly, attribution patching does not improve in LN-free models, suggesting its limitations stem from other nonlinearities. Finally, we showed that LN-free models lack operational entropy neurons, contributing towards the more generally observed trend of model overconfidence. More generally, LN-free models enable more precise mechanistic analyses when LN's nonlinearity becomes particularly problematic and, additionally, can serve as testbeds for evaluating the impact of the nonlinearities introduced by LN on mechanistic interpretability techniques. Open-sourcing these models, we hope to contribute to mechanistic interpretability research.

ACKNOWLEDGMENTS

JS and MS conducted this research as part of the MARS program by the Cambridge AI Safety Hub (CAISH). Furthermore, CAISH provided compute resources. LB and GK conducted this research as part of the Supervised Program for Alignment Research (SPAR). GK was supported by Open Philanthropy.

AUTHOR CONTRIBUTIONS

MS and JS worked jointly on the fine-tuning code and scaled up the LN-removal. MS contributed ideas behind auxiliary loss and recomputation and did attention sink rate analysis. JS optimized the removal schedule and recomputation. JS and MS conducted final experiments for the LN-free models reported in the manuscript.

LB and GK worked jointly on the mechanistic interpretability experiments. LB led the creation of the Pile-filtered dataset and the implementation of the attribution patching analysis. GK led the BOS token size and confidence neuron analyses. Both LB and GK contributed to the direct logit attribution experiments, with GK running the final experiments reported in the manuscript.

SH coordinated the project and provided advice and mentorship throughout.

All authors contributed to the writing.

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

# Appendix

## A  CODE AND MODEL AVAILABILITY

The LN removal code is available on GitHub `https://github.com/submarat/removing-layer-norm`

Table 2: Hugging Face links for models used and generated in this manuscript. Furthermore, fine-tuning (FT) steps and GPU hours are shown.

| Model | FT Steps | FT GPU Hours | Link |
|---|---|---|---|
| GPT-2 Small original | 0 | N/A | `https://huggingface.co/openai-community/gpt2` |
| GPT-2 Small vanilla | 300 | 1 | `https://huggingface.co/schaeff/gpt2-small_vanilla300` |
| GPT-2 Small LN-free | 300 | 1.5 | `https://huggingface.co/schaeff/gpt2-small_LNFree300` |
| GPT-2 Medium original | 0 | N/A | `https://huggingface.co/openai-community/gpt2-medium` |
| GPT-2 Medium vanilla | 500 | 2.5 | `https://huggingface.co/schaeff/gpt2-medium_vanilla500` |
| GPT-2 Medium LN-free | 500 | 3.5 | `https://huggingface.co/schaeff/gpt2-medium_LNFree500` |
| GPT-2 Large | 0 | N/A | `https://huggingface.co/openai-community/gpt2-large` |
| GPT-2 Large vanilla | 600 | 6.5 | `https://huggingface.co/schaeff/gpt2-large_vanilla600` |
| GPT-2 Large LN-free | 600 | 8 | `https://huggingface.co/schaeff/gpt2-large_LNFree600` |
| GPT-2 XL original | 0 | N/A | `https://huggingface.co/openai-community/gpt2-xl` |
| GPT-2 XL vanilla | 800 | 14 | `https://huggingface.co/schaeff/gpt2-xl_vanilla800` |
| GPT-2 XL LN-free | 800 | 26 | `https://huggingface.co/schaeff/gpt2-xl_LNFree800` |

**Other Compute Requirements:**  The evaluation and interpretability experiments require a negligible amount of compute (on the order of a few GPU hours).

## B  BLOCKWISE LN-REMOVAL SCHEDULES

All schedules use a warmup phase, cosine learning rate decay schedule, and continue fine-tuning for some iterations after LN removal is completed. Recomputation and auxiliary loss are applied to all schedules. The removal steps in the schedule are configured by start, gap and number of layers hyper parameters Tab. 3; See Tab. 4 for how these affect the final schedules.

| Hyperparameter | Small | Medium | Large | XL |
|---|---|---|---|---|
| Original GPT-2 model | gpt2 | gpt2-medium | gpt2-large | gpt2-xl |
| Micro Batch Size | 32 | 22 | 28 | 18 |
| Gradient Accumulation Steps | 16 | 23 | 18 | 28 |
| Batch Tokens Per Step | 524288 | 518144 | 516096 | 516096 |
| Weight Decay | 0.01 | 0.01 | 0.01 | 0.01 |
| Learning Rate | 0.0006 | 0.0006 | 0.0003 | 0.0001 |
| Min Learning Rate | 0.0003 | 0.0003 | 0.00004 | 0.00002 |
| Aux Loss Weight | 0.1 | 0.1 | 0.03 | 0.01 |
| Gradient Checkpointing | true | true | false | false |
| GPU memory | 80GB | 80GB | 180GB | 180GB |
| Number of Layers | 12 | 24 | 36 | 48 |
| Warmup Steps | 25 | 10 | 15 | 20 |
| Max Steps | 300 | 500 | 1200 | 1200 |
| Start $LN_{MLP}$ | 20 | 20 | 30 | 50 |
| Start $LN_{qk}$ | 44 | 68 | 174 | 242 |
| Start $LN_v$ | 68 | 116 | 318 | 434 |
| Start $LN_f$ | 104 | 188 | 534 | 722 |
| Gap $LN_{MLP}$ | 2 | 2 | 4 | 4 |
| Gap $LN_{qk}$ | 2 | 2 | 4 | 4 |
| Gap $LN_v$ | 3 | 3 | 6 | 6 |

Table 3: Comparison of GPT-2 Small, Medium, Large, and XL LN-free Hyperparameters

| | Small (12 layers) | | Medium (24 layers) | | Large (36 layers) | | XL (48 layers) | |
|---|---|---|---|---|---|---|---|---|
| | Step | Removal | Step | Removal | Step | Removal | Step | Removal |
| MLP | 20 | $LN_{MLP}^0$ | 20 | $LN_{MLP}^0$ | 30 | $LN_{MLP}^0$ | 50 | $LN_{MLP}^0$ |
| | 22 | $LN_{MLP}^1$ | 22 | $LN_{MLP}^1$ | 34 | $LN_{MLP}^1$ | 54 | $LN_{MLP}^1$ |
| | ... | ... | ... | ... | ... | ... | ... | ... |
| | 42 | $LN_{MLP}^{11}$ | 66 | $LN_{MLP}^{23}$ | 170 | $LN_{MLP}^{35}$ | 238 | $LN_{MLP}^{47}$ |
| QK | 44 | $LN_{qk}^0$ | 68 | $LN_{qk}^0$ | 174 | $LN_{qk}^0$ | 242 | $LN_{qk}^0$ |
| | 46 | $LN_{qk}^1$ | 70 | $LN_{qk}^1$ | 178 | $LN_{qk}^1$ | 246 | $LN_{qk}^1$ |
| | ... | ... | ... | ... | ... | ... | ... | ... |
| | 66 | $LN_{qk}^{11}$ | 114 | $LN_{qk}^{23}$ | 314 | $LN_{qk}^{35}$ | 430 | $LN_{qk}^{47}$ |
| V | 68 | $LN_v^0$ | 116 | $LN_v^0$ | 318 | $LN_v^0$ | 434 | $LN_v^0$ |
| | 71 | $LN_v^1$ | 119 | $LN_v^1$ | 324 | $LN_v^1$ | 440 | $LN_v^1$ |
| | ... | ... | ... | ... | ... | ... | ... | ... |
| | 101 | $LN_v^{11}$ | 185 | $LN_v^{23}$ | 528 | $LN_v^{35}$ | 716 | $LN_v^{47}$ |
| Final | 104 | $LN^f$ | 188 | $LN^f$ | 534 | $LN^f$ | 722 | $LN^f$ |

Table 4: LN removal schedule for GPT-2 Models (Small, Medium, Large, and XL). Values correspond to fine-tuning steps when a particular LN is removed. Gaps between removal events are uniform within each LN group.

### B.1 EMPIRICAL GUIDELINES FOR HYPERPARAMETER SELECTION

Based on our empirical observations during the development of the LN-removal procedure, we identified several common failure modes and here we provide effective mitigation strategies:

**Exploding loss after removal events:** If the main loss spikes dramatically shortly after removing a LN module, this typically indicates that the removal schedule is too aggressive. The most effective mitigation is to increase the gaps between removal events within each LN group.

**Loss degradation after complete LN removal:** When the loss begins to increase after all LN modules have been removed, we found it beneficial to increase the minimum learning rate of the learning rate schedule.

**Sudden failures at some fine-tuning step during the query/key path LN removal schedule:** In this specific case, we found it beneficial to monitor standard deviation values for drop in replacement, modify the EMA smoothing factor, change the learning rate, and change the auxiliary loss weight.

These strategies proved effective for achieving stable LN removal across all model sizes. Additionally, the auxiliary loss and EMA estimation significantly helped in reducing sensitivity to hyperparameter choices compared to approaches without these components.

### B.2 SUBLINEAR SCALING OF FINETUNING DATA REQUIREMENTS

Analyzing the total number of tokens used during LN removal across different GPT-2 variants (Fig. 1 and Table 3), it is possible to observe sublinear scaling of fine-tuning data requirements with model size. From our results:

- GPT-2 Small (124M params): 157M tokens ( 300 steps × 524k tokens/step)
- GPT-2 XL (1.5B params): 413M tokens ( 800 steps × 516k tokens/step)

This shows a 12× increase in parameters required only a 2.6× increase in fine-tuning tokens.

## C GENERALIZATION BEYOND GPT-2 FAMILY

To evaluate whether our LN removal methodology extends beyond the GPT-2 model family, we conducted additional experiments using Pythia 70M (Biderman et al., 2023). We compared two fine-tuning approaches on The Pile dataset: vanilla fine-tuning with LN intact, and our proposed fine-tuning strategy where LN is removed.

The results demonstrate comparable performance degradation to our GPT-2 findings. The vanilla fine-tuned model achieved a cross-entropy loss of 3.80 on the OpenWebText evaluation set, while the LN-free variant achieved a cross-entropy loss of 3.89. This 0.09 increase in loss aligns closely with the performance gap observed in GPT-2 Small (124M parameters), indicating that our methodology exhibits consistent behavior across different transformer architectures. On the HellaSwag benchmark, the original Pythia 70M model scored 0.2679, the vanilla fine-tuned model achieved 0.2667, and the LN-free model obtained 0.2636, showing minimal performance degradation on this alternative evaluation.

These findings provide evidence that our LN removal approach generalizes beyond the specific GPT-2 architecture to other transformer-based language models.

## D  RELATIONSHIP BETWEEN LN AND RMSNORM AND GENERALIZATION TO RMSNORM MODELS

RMSNorm is defined as

$$\text{RMSNorm}(\mathbf{x}) = \frac{\mathbf{x}}{\sigma} \odot \boldsymbol{\gamma}, \quad \sigma = \sqrt{\frac{1}{H} \sum_{h=1}^{H} (x_h)^2}. \tag{8}$$

As such LN and RMSNorm are nearly equivalent operations, differing solely in three aspects:

- LayerNorm includes a bias term $\boldsymbol{\beta}$, which can however be "folded" (i.e., absorbed into the parameters of an adjacent linear operation).
- Subtracting $\mu$ before rescaling, LayerNorm effectively removes the $(1, 1, 1, \ldots)$ component of the activation vector.
- LayerNorm rescales by the standard deviation of the activation component values rather than the RMS. However, after removing the $(1, 1, 1, \ldots)$ projection, rescaling by the standard deviation is equivalent to rescaling by the RMS up to a factor of $\sqrt{d_{\text{model}}}$, which can as well be "folded" into adjacent parameters.

Since LayerNorm is applied before every model component and before the unembedding, the net effect is that LayerNorm models do not utilize the $(1, 1, 1, \ldots)$ dimension of their residual stream and apply RMS normalization on the remaining dimensions. For further discussion, see Gupta et al. (2025). Given this near-equivalence, we expect our procedure to generalize to models using RMSNorm.

# E   THE PILE-FILTERED

When evaluating models on the Pile (Gao et al., 2020),[6] we observed unusually high cross-entropy losses for specific tokens. To investigate this, we compared token frequency distributions between 1 million samples from this dataset and OpenWebText (Gokaslan & Cohen, 2019),[7] both pretok-enized with GPT-2. We identified tokens that appeared in The Pile but not in OpenWebText, which corresponded to sequences with high cross entropy loss. We filtered out sequences containing any of these tokens, and created a small 10,000-example filtered subset of The Pile.   The filtered dataset, along with token metadata and generation scripts, is made on the Hugging Face Hub `https://huggingface.co/datasets/lucabaroni/apollo-pile-filtered-10k`.

| token_id | token | count |
|---------:|-------|------:|
| 197 | \t | 4,260,185 |
| 628 | \n\n | 1,382,601 |
| 1849 | \xa0 | 1,090,135 |
| 201 | \r | 725,891 |
| 191 | \x03 | 50,457 |
| 200 | \x0c | 49,412 |
| 5624 | \xa0 | 40,045 |
| 4603 | \xa0\xa0 | 9,374 |
| 205 | \x11 | 5,169 |
| 203 | \x0f | 4,177 |

Table 5: Top 10 most frequent tokens present in The Pile and missing in OpenWebText.

## E.1   GPT-2 XL LN-FREE HIGH LOSS SAMPLES ON THE PILE

We reported a very high mean CE loss (130.22) for GPT-2 XL LN-Free on The Pile.  However, the median and 99.9 percentile range are very similar to GPT-2 XL original.  Three samples are responsible for the high mean CE loss for GPT-2 XL LN-Free on The Pile. We list these samples below. These samples contain a token or token sequence not present in OWT and are listed in Tab. 5. At such tokens, the model has absurdly high CE losses, up to 5 million, i.e., the model is overconfident that the true next token will not be the next. For the three samples, the first token prediction with CE loss larger than 50 are "\x0c", "\t", and "\n" respectively. The last token of the sequence leading up to the token with high errors is "\n" for all three samples, indicating that these specific tokens and token combinations are causing overconfidence in the model. Further inspection reveals that these high CE losses derive from very large negative logits. These outliers occur because, in rare cases involving certain tokens not present in the fine-tuning dataset, the norm of residual stream vectors before unembedding explodes. Interestingly, we observed this phenomenon only in GPT-2 XL.

**Sample 1:**

```
Sample 2726 out of 10k has tokens with CE loss > 50.

First token with CE loss > 50:200  at position 11.
Decoded:'
'
Decoded (unicode_escape):'\x0c'
Sequence of last 5 Tokens for prediction:220 220 220 1367 198
Decoded:'   11
'
Decoded (unicode_escape):'   11\n'

(Token:Loss)
220:N/A, 220:7.6214733, 220:7.988017, 220:0.7575181, 220:0.21067815, 220:0.11241462, 220:0.0828728, 220:0.07294927,
220:0.06983218, 1367:9.5656, 198:3.8515434, 200:54.273285, 42138:11.611183, 290:5.8352313, 2912:9.315803, 9021:17.121414,
286:5.927439, 8460:9.354071, 642:3.973015, 4310:4.678949, 761:10.288656, 407:0.60814863, 307:0.55970573, 3940:4.7689095,
```

---

[6]Specifically   we   used   `https://huggingface.co/datasets/apollo-research/monology-pile-uncopyrighted-tokenizer-gpt2`
[7]Specifically   we   used   `https://huggingface.co/datasets/apollo-research/Skylion007-openwebtext-tokenizer-gpt2`

13:1.346435, 41990:9.208092, 2173:7.9507837, 503:0.88767886, 326:0.47304547, 287:4.0013585, 428:2.462367, 198:7.3526363, 198:0.0011684026, 7442:1.6571776, 11:0.8788041, 262:1.2546973, 20693:6.1989183, 4934:4.3370743, 284:2.2307296, 38040:3.9622679, 10494:0.005666858, 19303:9.20058, 2457:4.18554, 3173:1.1498255, 1682:8.386018, 2058:4.2735405, 407:3.1076946, 422:0.1769652, 262:0.78079456, 198:2.1329598, 198:0.00015055, 36208:13.958086, 4537:12.094295, 16412:8.990057, 475:3.527892, 422:0.22476129, 262:0.55110574, 3893:8.833248, 17541:5.443501, 4347:10.620082, 1799:5.924607, 290:3.480315, 37159:10.124819, 2191:5.4837275, 286:1.5630095, 8235:9.437175, 357:4.602768, 447:6.1605263, 250:5.47846, 39:5.972879, 4061:5.5613327, 3838:6.2311015, 447:4.11596, 251:0.2733165, 828:5.8963223, 198:3.1704738, 198:3.05913, 14876:12.424786, 13:2.642362, 406:8.8554945, 13:4.501826, 1400:6.3063745, 13:2.2849065, 14436:11.087215, 12:3.5680172, 26492:13.83733, 11:2.9488518, 47171:9.930283, 8949:10.020365, 11:2.4528143, 15143:9.181636, 11:2.0323732, 22219:9.782476, 11:1.5124732, 9796:9.335302, 5133:8.786331, 13:2.5590122, 27653:13.353212, 11:1.7378397, 27937:10.412004, 11:1.2108434, 15408:11.124487, 11:1.0378554, 1160:8.236352, 6469:10.613097, 357:3.7660804, 22288:7.9838486, 828:5.7518673, 543:3.3631327, 198:4.0904975, 198:0.05241805, 1939:11.162535, 40132:2.9781475, 1115:7.43649, 13788:11.595028, 10411:5.7753525, 8617:7.3080463, 656:7.60578, 13793:10.331831, 22312:10.301449, 11:1.4195569, 262:2.8994842, 18628:10.344179, 20197:7.251051, 6127:7.2656045, 11:2.1663508, 290:2.5125058, 198:4.6340384, 198:2.3188238, 1169:6.9466467, 5094:8.366037, 3893:6.32104, 4809:6.82047, 2191:6.6650662, 25:4.863468, 628:8.807043, 220:8.19655, 220:3.819473, 220:4.2062297, 220:4.961961, 220:4.995181, 220:5.277912, 383:5.82012, 4986:9.652403, 11:2.5059075, 6414:9.438324, 351:3.5732212, 2665:7.6105623, 14436:8.626756, 286:4.936455, 262:2.6958165, 3893:9.572085, 7276:6.6044493, 4347:11.077166, 1799:6.549804, 290:4.471074, 14487:11.074, 220:5.064687, 220:5.9405313, 220:4.124799, 220:3.2302897, 220:1.6753389, 37159:11.453347, 2191:6.410646, 286:3.9341471, 8235:10.6037035, 11:2.1030743, 743:7.7276053, 38040:13.952975, 10494:12220.764, 884:1568.6077, 6647:2484.7053, 355:883.05884, 743:1939.9294, 307:1384.4974, 3306:2680.565, 198:469.1405, 220:1538.7728, 220:1178.9397, 220:1431.203, 220:1082.1162, 220:1259.7878, 220:1304.2883, 393:397.98828, 5035:2698.9102, 284:456.9895, 3283:2395.4785, 503:1655.8503, 262:674.69543, 8617:3220.739, 286:830.44946, 428:1525.5525, 685:797.9059, 3911:4231.721, 4083:2158.9766, 383:999.2892, 4986:2601.727, 743:1726.6406, 198:450.30127, 220:1403.6172, 220:2175.758, 220:2173.087, 220:1988.6704, 220:1560.3451, 220:1146.8448, 38040:4204.649, 10494:8649.862, 597:1345.0178, 19303:3773.4775, 2457:2029.5072, 3173:1839.6007, 355:568.34094, 262:590.8824, 4986:2062.4531, 15947:2279.8892, 389:1038.8564, 5035:2869.369, 284:779.21716, 198:396.0829, 220:881.5056, 220:1608.5259, 220:2106.2659, 220:1546.5009, 220:1546.5249, 220:1340.715, 3283:2450.9758, 503:1587.5262, 4347:1318.5421, 685:882.85913, 3911:3507.6172, 4083:2057.4734, 198:216.41724, 198:246.9541, 1959:2581.9805, 471:1004.4808, 13:41.873535, 50:1349.8, 13:150.74365, 34:1408.8376, 13:197.64563, 8460:2291.2534, 15136:1785.7683, 16:2138.023, 66:1901.7491, 11:178.7539, 2608:2297.8555, 471:1437.7157,

...

Decoded:

11

notice and comment procedures of § 553 need not be followed. Plaintiff points out that in this case, the statutory authority to promulgate interim final rules actually comes not from the MHPAEA but from the Health Insurance Portability and Accountability Act of 1996 ("HIPAA"), Pub. L. No. 104–191, §§ 101, 102, 401, 110 Stat. 1936, 1951, 1976, 2082 (1996), which incorporated three substantially identical provisions into ERISA, the Internal Revenue Code, and the Public Health Service Act:

> The Secretary, consistent with section 104 of the Health Care Portability and Accountability Act of 1996, may promulgate such regulations as may be necessary or appropriate to carry out the provisions of this [part]. The Secretary may promulgate any interim final rules as the Secretary determines are appropriate to carry out this [part].

29 U.S.C. § 1191c, 26 U.S.C. § 9833 (replacing "part" with "chapter"), and 42 U.S.C. § 300gg-92 (replacing "part" with "subchapter").4 Plaintiff argues that Congress only intended to give the Secretaries authority to promulgate interim final rules relating to HIPAA and not the MHPAEA, which was passed twelve years later. However, the MHPAEA's substantive provisions are amendments to the same sections of ERISA, the Internal Revenue Code, and the Public Health Service Act that are governed by the HIPAA provisions cited above, and the statutory text clearly gives the Secretaries authority to promulgate interim final rules to carry out these sections. Therefore, the Court finds that Congress has authorized the Secretaries to "promulgate any interim final rules as the[y] determine[] are appropriate to carry out the" MHPAEA.

Finding that Congress authorized the promulgation of interim final rules does not end the inquiry. Although the APA recognizes that Congress may modify the notice and comment

4

This regulatory authority covers part 7 of Subtitle B of Title I of ERISA (29 U.S.C. §§ 1181-91c), Chapter 100 of the Internal Revenue Code (26 U.S.C. §§ 9801-33), and Part A of Title XXVII of the Public Health Service Act (42 U.S.C. §§ 300gg to 300gg-92).

12

procedures called for by § 553, it states that a "[s]ubsequent statute may not be held to supersede or modify [§ 553] . . . except to the extent that it does so expressly." 5 U.S.C. § 559. "[T]he import of the § 559 instruction is that Congress's intent to make a substantive change be clear." Ass'n of Data Processing Serv. Orgs., Inc. v. Bd. of Governors, 745 F.2d 677, 686 (D.C. Cir. 1986). The statutory provisions authorizing interim final rules in this case do not mention notice

and comment or any other aspect of the APA. In such a case, the D.C. Circuit has defined the

relevant standard as "whether Congress has established procedures so clearly different from those

required by the APA that it must have intended to displace the norm." Asiana Airlines v. FAA,

134 F.3d 393, 397 (D.C. Cir. 1998).

    Defendants rely on two cases in which the D.C. Circuit held that the notice and comment

provisions of § 553 were abrogated by specific statutory provisions authorizing interim final

rules. See Asiana Airlines v. Fed. Aviation Admin., 134 F.3d 393 (D.C. Cir. 1998); Methodist

Hosp. of Sacramento v. Shalala, 38 F.3d 1225 (D.C. Cir. 1994). In Methodist Hospital of

Sacramento, the court was faced with

## Sample 2:

```
Sample 7323 out of 10k has tokens with CE loss > 50.
First token with CE loss > 50:197  at position 10.
Decoded:' '
Decoded (unicode_escape):'\t'
Sequence of last 5 Tokens for prediction:257 4731 7177 13 198
Decoded:' a string array.
'
Decoded (unicode_escape):' a string array.\n'
(Token:Loss)
1003:N/A, 1003:11.26658, 9726:11.162002, 46621:13.591053, 355:5.1217504, 257:1.9146276, 4731:7.4259768, 7177:8.642193,
13:1.57043, 198:1.4427543, 197:51.156723, 12235:14.619279, 39:10.696115, 7465:9.209376, 17635:11.363218, 8841:10.934766,
198:3.474833, 92:13.564951, 198:1.2806269, 198:0.0055186776, 1003:3.836342, 968:7.883165, 49:8.290166, 3798:5.798291,
272:7.7695932, 45356:11.567278, 40109:11.504518, 5860:9.337919, 257:1.5190241, 649:1.7121754, 4554:2.7947285, 543:4.841503,
460:2.3216853, 307:0.686766, 973:0.74536353, 284:0.4934966, 2071:5.992473, 257:1.6754444, 581:7.8475504, 5171:6.433069,
198:3.4856787, 1003:13.67105, 19449:8.2021055, 12:3.5855105, 49:5.30799, 5662:5.4760623, 3141:7.5336666, 13:2.6725016,
198:0.8083212, 1003:15.782787, 198:2.8173897, 1003:15.395724, 24550:5.2747335, 25:0.19102867, 770:2.1424239, 318:1.7225417,
257:1.5685425, 275:8.388821, 83:4.2324853, 10210:5.020052, 7552:5.175566, 49702:11.925024, 422:0.8439282, 33084:4.970866,
13:0.68376416, 785:0.20635764, 14:0.37836862, 12501:8.026502, 445:2.2101464, 14:0.4466647, 17896:7.95566, 4372:3.105786,
14:2.777247, 67:4.3537035, 6098:0.37673652, 17752:5.617012, 198:2.2561002, 1003:12.216859, 290:5.0420575, 4433:4.454626,
257:1.9344062, 2639:6.557314, 5459:0.1832912, 4637:3.1329556, 13:1.564306, 198:0.2661691, 20786:21.258528, 968:1.6345162,
49:0.024241818, 3798:2.4097002, 272:3.8780181, 45356:10.635372, 40109:9.851566, 7:3.763564, 9967:6.12138, 39:6.5091047,
7465:0.047564577, 17635:5.3687067, 8841:1.8500897, 8:6.0806694, 1635:5.2731657, 49:4.241615, 3798:2.2475796, 272:1.7413952,
45356:6.1853223, 40109:9.715792, 1391:10.957037, 198:3.2365587, 197:35.354282, 7783:13.6022215, 1222:7.565274, 49:5.145051,
3798:12.716011, 272:9.562733, 45356:12.930087, 40109:13.774559, 90:9.70529, 12235:5.3254843, 39:13.270555, 7465:12.182639,
25:3.5354931, 2512:9.293187, 39:12.1883745, 7465:14.035143, 92:11.704035, 198:3.7612562, 92:9.557363, 198:4.632967,
198:1.6126469, 20786:11.7582035, 2315:11.252395, 3419:599915.1, 1391:128406.42, 198:21275.512, 197:436295.1, 1003:167360.28,
383:44864.34, 9729:140597.66, 287:10736.5625, 428:79266.625, 2393:104939.37, 389:51231.992, 691:73092.914, 24284:177729.25,
416:58283.36, 2639:149387.14, 11603:219064.45, 13:6765.0703, 198:32484.742, 197:443424.3, 33152:201537.97, 19039:195013.5,
471:62431.207, 37:97675.016, 1135:110187.266, 1443:234881.84, 5459:269756.94, 10049:242653.28, 628:112184.06, 197:436574.7,
34320:148946.5, 38804:172249.12, 40109:219531.69, 7203:188346.6, 41299:237815.0, 5344:139417.62, 1600:119173.45, 20789:137373.7,
47649:202613.36, 5344:148038.86, 40109:232167.31, 5769:212062.88, 45991:286692.56, 828:95972.62, 9701:161537.31,
8:45388.77, 198:26669.29, 197:461318.66, 34320:180015.84, 38804:183880.8, 40109:273160.8, 7203:128039.91, 2220:195715.38,
17602:179212.78, 24455:165867.17, 1600:177894.75, 20789:196518.0, 8912:206497.88, 46047:205311.12,
22417:217080.81, 40109:232314.2, 5769:212559.25, 45991:271713.0, 828:92336.35, 9701:131772.14, 8:71338.83, 198:9289.922,
197:479748.06, 34320:161961.17, 38804:183598.38, 40109:324803.62, 7203:168302.4, 1662:152243.08, 1958:311552.6,
27372:174300.81, 1600:175331.19, 20789:198232.06, 3673:121481.22, 1958:134500.3, 45356:281647.62, 40109:218485.22,

...

Decoded:
 // Block hashes as a string array.
 BlockHashes []string
}

// NewRescanBlocksCmd returns a new instance which can be used to issue a rescan
// JSON-RPC command.
//
// NOTE: This is a btcd extension ported from github.com/decred/dcrd/dcrjson
// and requires a websocket connection.
func NewRescanBlocksCmd(blockHashes []string) *RescanBlocksCmd {
 return &RescanBlocksCmd{BlockHashes: blockHashes}
}

func init() {
 // The commands in this file are only usable by websockets.
 flags := UFWebsocketOnly

 MustRegisterCmd("authenticate", (*AuthenticateCmd)(nil), flags)
 MustRegisterCmd("loadtxfilter", (*LoadTxFilterCmd)(nil), flags)
 MustRegisterCmd("notifyblocks", (*NotifyBlocksCmd)(nil), flags)
 MustRegisterCmd("notifynewtransactions", (*NotifyNewTransactionsCmd)(nil), flags)
 MustRegisterCmd("notifyreceived", (*NotifyReceivedCmd)(nil), flags)
 MustRegisterCmd("notifyspent", (*NotifySpentCmd)(nil), flags)
 MustRegisterCmd("session", (*SessionCmd)(nil), flags)
 MustRegisterCmd("stopnotifyblocks", (*StopNotifyBlocksCmd)(nil), flags)
 MustRegisterCmd("stopnotifynewtransactions", (*StopNotifyNewTransactionsCmd)(nil), flags)
 MustRegisterCmd("stopnotifyspent", (*StopNotifySpentCmd)(nil), flags)
 MustRegisterCmd("stopnotifyreceived", (*StopNotifyReceivedCmd)(nil), flags)
 MustRegisterCmd("rescan", (*RescanCmd)(nil), flags)
 MustRegisterCmd("rescanblocks", (*RescanBlocksCmd)(nil), flags)
```

```
}
Faithless Execution: Fighting Presidential Lawlessness

The first few days of rolling out my new book, Faithless Execution, have been exhilarating, with few things more
gratifying and humbling than the wonderful review by one of my very favorites, PJ Media's own Roger Simon.

It has been uplifting to see how many people really are alarmed--rather than indifferent, as I worried--to the problem of rampant
presidential lawlessness. People really do grasp that the separation of powers, which is so threatened by President
Obama's usurpation of the powers of the states and other federal departments, really is the key to protecting our liberties. Too
much accumulation of power in one government official's hand--particularly, the Framers observed, the joining of the
legislative and executive power in a single department or person--is the road to tyranny.

When people grasp that, they similarly grasp that presidential lawlessness is not a conservative versus liberal issue, nor
Republican versus Democrat. It is a question of whether we still aspire to be a republic under the rule of law instead of
subjects under presidential whim. If they are not knocked down, the precedents that President Obama is setting for imperial
executive power will be available for exploitation by every future president, regardless or party or ideological
orientation. That ought to frighten all Americans, not just opponents of the current president's policies.

I make a sustained attempt in the book to explain that impeachment--the ultimate constitutional response to presidential
lawlessness--is a political remedy, not a legal one. You can have a thousand impeachable offenses, but if there is not a strong
public will that the president be removed, impeachment is a nonstarter. The political case for removal is the one that is
uphill. Establishing the legal case for impeachment--i.e., demonstrating that high crimes and misdemeanors have been
committed--is the easy part.// The label and actions expect to be in a flex container. Since this component adds another

// wrapping layer to the mdc-snackbar__surface, it should also include flex display.
.mat-mdc-simple-snack-bar {
  display: flex;
}

"It was like the Alamo at times. Nothing went for us. It feels like we have lost but the final is over two legs and we have to
be delighted with the overall scoreline."

Liverpool first-team captain Steven Gerrard and central defender Jamie carragher were quickly in touch after the win and Heighway
added: "They have followed us all the way through.

"They texted us before every game and they have texted us again after the win.

"They are steeped in the history of this club and know what it means to win this tournament."

City's academy chief Jim Cassell
```

## Sample 3:

```
Sample 9335 out of 10k has tokens with CE loss > 50.
First token with CE loss > 50:198  at position 155.
Decoded:'
'
Decoded (unicode_escape):'\n'
Sequence of last 5 Tokens for prediction:49704 49704 9705 20379 198
Decoded:'////////////////////////////////////////////////////////////////////
'
Decoded (unicode_escape):'////////////////////////////////////////////////////////////////////////////\n'
(Token:Loss)
407:N/A, 407:4.6831055, 1624:7.360232, 326:1.1473916, 345:3.3804011, 2630:7.6741157, 262:1.3505429, 2656:4.104636,
3788:4.6768866, 13:1.0597951, 1002:2.57059, 345:0.35921186, 779:3.5662773, 428:2.6555552, 3788:0.4169199, 287:1.3137746,
257:0.32647714, 1720:1.4579158, 11:0.9270898, 281:3.5136762, 48182:1.3712287, 287:0.2596935, 262:0.058141652,
1720:0.16443609, 10314:0.29576224, 561:0.5035581, 307:0.02398988, 16373:0.48682904, 475:0.81268287,
318:0.05656958, 407:0.007545187, 2672:0.050823122, 13:0.0214831, 198:0.7484702, 17:13.849314, 13:0.13205929, 978:7.044759,
4400:2.0608654, 2723:2.2134705, 6300:2.333941, 1276:0.88757795, 307:0.4205811, 30723:1.4280686, 7498:0.11422959,
355:0.0134238945, 884:0.00481102, 11:0.5958381, 290:0.08005254, 1276:0.2439856, 407:0.08167637, 307:0.025275672,
26521:0.17200725, 276:0.0023700502, 355:0.0029704517, 852:0.14899838, 262:0.0020197486, 2656:0.022728885,
3788:0.20550326, 13:0.04337017, 198:0.31131023, 18:6.3787313, 13:0.00059801334, 770:0.9271791, 4003:1.7363278, 743:1.0793377,
407:0.281329, 307:0.013555973, 4615:0.9492111, 393:0.0392538, 14294:0.19510294, 422:0.03199716, 597:0.038671132,
2723:0.24237014, 6082:0.28850555, 13:0.014584245, 198:0.0928015, 16208:7.4476423, 198:0.10754685, 198:0.00033825875,
49704:7.4027767, 49704:0.059470795, 49704:2.1608517, 49704:1.4772909, 49704:1.1133443, 49704:0.9488324,
9705:2.1151383, 20379:1.1406435, 198:0.10158871, 35343:20.283905, 198:0.5157568, 1635:10.543703, 197:30.43997,
4264:13.305223, 1299:0.07563411, 2438:3.0418704, 329:1.6119438, 281:4.073881, 317:5.7669916, 6242:7.9582887, 33:0.08353172,
2927:11.291942, 1304:2.9536972, 13:1.0874708, 198:1.0187862, 1635:12.628989, 197:33.37102, 59:6.926921, 7753:7.370697,
197:33.560585, 197:31.852251, 3185:14.746413, 34:4.736884, 62:2.4475694, 3838:5.085874, 33833:6.238099, 692:5.69899,
1304:0.86001503, 13:0.34363738, 71:1.2452692, 198:0.81940943, 1635:9.853174, 197:32.87166, 59:3.7674747, 9800:6.4635477,
197:33.271984, 197:31.412739, 36910:15.742162, 3813:8.258679, 67:5.3074374, 24086:2.1277742, 198:0.7815698, 1635:3.943909,
197:33.18875, 59:9.100214, 4475:15.064762, 197:33.80469, 197:38.404465, 21339:16.17667, 11:3.2231097, 352:6.5712805,
301:12.503309, 11:3.5788884, 6244:13.497032, 198:4.3184443, 9466:12.119115, 198:4.7258415, 49704:23.561052, 49704:21.665047,
49704:19.027508, 49704:20.011747, 49704:21.655386, 49704:40.67674, 9705:22.09228, 20379:27.61382, 198:8.27015,
198:399.37866, 49704:809.2328, 49704:763.74054, 49704:722.9032, 49704:811.80115, 49704:693.3036, 49704:730.6386, 9705:677.8175,
20379:544.15265, 198:84.33852, 1003:409.64636, 40348:589.6119, 4932:453.42014, 198:66.265816, 2:368.93063, 361:344.89172,
358:423.16663, 891:447.8988, 11593:460.6718, 3185:463.0637, 34:137.90085, 62:323.4264, 3838:367.394, 33833:424.9605,
46:197.75824, 3069:536.48846, 41237:674.64325, 62:188.37312, 39:61808.293, 834:1737813.5, 198:58819.594, 2:501469.34,
13086:476518.62, 11593:430323.34, 3185:436209.5, 34:149311.72, 62:225198.66, 3838:499536.16, 33833:440378.56, 46:344437.5,
3069:506636.12, 41237:671244.25, 62:330165.34, 39:305198.0, 834:412493.72, 628:283896.75, 197:1610916.9, 7249:487438.94,
440:274647.84, 5662:508110.75, 16820:601000.5, 62:200508.23, 17614:568054.25, 317:109115.234, 6242:376880.62, 2749:538623.9,
4891:787976.1, 1058:368174.3, 14701:435548.7, 30562:738873.94, 198:122744.25, 197:1227677.4, 90:353759.03, 198:107419.83,
197:1258664.6, 197:1695466.2, 197:1546377.4, 197:1209663.6, 197:1201286.0, 197:1118433.5, 3838:524715.4, 33833:448043.62,
4891:523260.6, 3419:404699.94, 1058:314273.38, 12301:389488.12,

....

Decoded:
 must not claim that you wrote the original software. If you use this software in a product, an acknowledgment in the product
 documentation would be appreciated but is not required.
2. Altered source versions must be plainly marked as such, and must not be misrepresented as being the original software.
```

```
3. This notice may not be removed or altered from any source distribution.
*/

///////////////////////////////////////////////////////////////////////////////////////////////////////////////////
///////////////////////////////////////////////////////////////////////
/**
 * Contains code for an AABB collider.
 * \file   OPC_AABBCollider.h
 * \author Pierre Terdiman
 * \date   January, 1st, 2002
 */
///////////////////////////////////////////////////////////////////////////////////////////////////////////////////
///////////////////////////////////////////////////////////////////////

///////////////////////////////////////////////////////////////////////////////////////////////////////////////////
///////////////////////////////////////////////////////////////////////
// Include Guard
#ifndef __OPC_AABBCOLLIDER_H__
#define __OPC_AABBCOLLIDER_H__

 struct OPCODE_API AABBCache : VolumeCache
 {
      AABBCache() : FatCoeff(1.1f)
      {
       FatBox.mCenter.Zero();
       FatBox.mExtents.Zero();
      }

 // Cached faces signature
 CollisionAABB FatBox;  //!< Box used when performing the query resulting in cached faces
 // User settings
 float  FatCoeff; //!< mRadius2 multiplier used to create a fat sphere
 };

 class OPCODE_API AABBCollider : public VolumeCollider
 {
 public:
 // Constructor / Destructor
          AABBCollider();
 virtual        ~AABBCollider();

 ///////////////////////////////////////////////////////////////////////////////////////////////////////
        ///////////////////////////////////////////////////////////////////////////////
 /**
  * Generic collision query for generic OPCODE models. After the call, access the results:
  * - with GetContactStatus()
  * - with GetNbTouchedPrimitives()
  * - with GetTouchedPrimitives()
  *
  * \param  cache    [in/out] a box cache
  * \param  box      [in] collision AABB in world space
  * \param  model    [in] Opcode model to collide with
  * \return  true if success
  * \warning SCALE NOT SUPPORTED. The matrices must contain rotation & translation parts only.
  */
 ///////////////////////////////////////////////////////////////////////////////////////////////////////
        ///////////////////////////////////////////////////////////////////////////////
      bool  Collide(AABBCache& cache, const CollisionAABB& box, const Model& model);
 //
      bool  Collide(AABBCache& cache, const CollisionAABB& box, const AABBTree* tree);
 protected:
      CollisionAABB mBox;   //!< Query box in (center, extents) form
      Point  mMin;   //!< Query box min point
      Point  mMax;   //!< Query box max point
 // Leaf description
      Point  mLeafVerts[3]; //!< Triangle vertices
 // Internal methods
      void  _Collide(const AABBCollisionNode* node);
      void  _Collide(const AABBNoLeafNode* node);
      void  _Collide(const AABBQuantizedNode* node);
      void  _Collide(const AABBQuantizedNoLeafNode* node);
      void  _Collide(const AABBTreeNode* node);
      void  _CollideNoPrimitiveTest(const AABBCollisionNode* node);
      void  _CollideNoPrimitiveTest(const AABBNoLeafNode* node
```

## F   EVALUATION ON STANDARD BENCHMARKS

We additionally evaluated our models on four widely used benchmarks: BoolQ (Clark et al., 2019), HellaSwag (Zellers et al., 2019), PIQA (Bisk et al., 2020), and WinoGrande (Sakaguchi et al., 2021). These tasks assess general language understanding, commonsense reasoning, and pronoun resolution. Tables 6–9 report normalized accuracy for each model before and after LayerNorm removal. LN-free models maintain performance comparable to their baselines, with only minor variations across tasks.

| Task | GPT-2 XL original | GPT-2 XL vanilla FT | GPT-2 XL LN-free FT |
|------|-------------------|---------------------|---------------------|
| BoolQ | 61.8 | **62.2** | 61.9 |
| HellaSwag | **50.9** | 49.8 | 48.8 |
| PIQA | **70.5** | 70.5 | 69.9 |
| WinoGrande | **58.3** | 57.5 | 56.1 |

Table 6: Accuracy of GPT-2 XL model variants on BoolQ, HellaSwag, PIQA, and WinoGrande.

| Task | GPT-2 Large original | GPT-2 Large vanilla FT | GPT-2 Large LN-free FT |
|------|----------------------|------------------------|------------------------|
| BoolQ | 60.5 | **62.1** | 62.0 |
| HellaSwag | **45.4** | 43.4 | 42.8 |
| PIQA | 69.2 | 68.7 | **69.3** |
| WinoGrande | 55.3 | **56.2** | 54.6 |

Table 7: Accuracy of GPT-2 Large model variants on BoolQ, HellaSwag, PIQA, and WinoGrande.

| Task | GPT-2 Medium original | GPT-2 Medium LN-free FT |
|------|-----------------------|-------------------------|
| BoolQ | 58.6 | **59.9** |
| HellaSwag | **39.4** | 37.4 |
| PIQA | **66.4** | 65.6 |
| WinoGrande | **53.1** | 51.5 |

Table 8: Accuracy of GPT-2 Medium model variants on BoolQ, HellaSwag, PIQA, and WinoGrande.

| Task | GPT-2 Small original | GPT-2 Small LN-free FT |
|------|----------------------|------------------------|
| BoolQ | 48.7 | **52.0** |
| HellaSwag | **31.1** | 30.2 |
| PIQA | **62.5** | 61.4 |
| WinoGrande | **51.6** | 50.9 |

Table 9: Accuracy of GPT-2 Small model variants on BoolQ, HellaSwag, PIQA, and WinoGrande.

Overall, these results show that LN-free models maintain comparable performance on standard benchmarks, supporting their use in interpretability studies.

## G    DIRECT LOGIT ATTRIBUTION BECOMES EXACT

To quantify the discrepancy between Direct Logit Attribution (DLA) and the Direct Effect (DE), we compute the Normalized Mean Absolute Error (NMAE) through a two-stage process. For each attention head $h$ in the model, we calculate:

$$\text{NMAE}_h = \frac{\frac{1}{N}\sum_{i=1}^{N}|DLA_{i,h} - DE_{i,h}|}{\frac{1}{N}\sum_{i=1}^{N}|DE_{i,h}|} \times 100\% \tag{9}$$

where, $i$ indexes individual sequences in The Pile-Filtered dataset, $h$ indexes attention heads, $DLA_{i,h}$ and $DE_{i,h}$ are the corresponding DLA and DE values for sequence $i$ and head $h$. We then average across all attention heads to obtain the overall NMAE:

$$\text{NMAE} = \frac{1}{N_{heads}} \sum_{h=1}^{N_{heads}} \text{NMAE}_h \tag{10}$$

The original model exhibits an NMAE of 49.07% [29.92%, 66.10%] (95% Confidence Interval - CI), indicating that Direct Linear Attribution (DLA) estimates deviate from direct effect measurements by approximately half of the true effect magnitude on average across all attention heads. The vanilla fine-tuned model demonstrates an even larger discrepancy with an NMAE of 57.85% [38.52%, 74.52%]. In contrast, the LN-free fine-tuned model achieves a perfect 0.00% [0.00%, 0.00%] NMAE, empirically confirming that removing the non-linearity introduced by LN eliminates the discrepancy between DLA and direct ablation methods. This result validates that without LN's non-linearity, the two attribution methods are mathematically equivalent, eliminating the need for linearization approximations, which can be significantly inaccurate.

To ensure our overall NMAE metric was not biased by a few significant outliers, we visualized the per-head NMAE values across all attention heads in GPT-2 Small models (Figure 5). In models with LN, the disagreement is widespread across most attention heads rather than driven primarily by a small number of outliers. Later layers showed NMAE values exceeding 100%. In contrast, the LN-free model shows no disagreement between both methods, with NMAE values of zero across all heads.

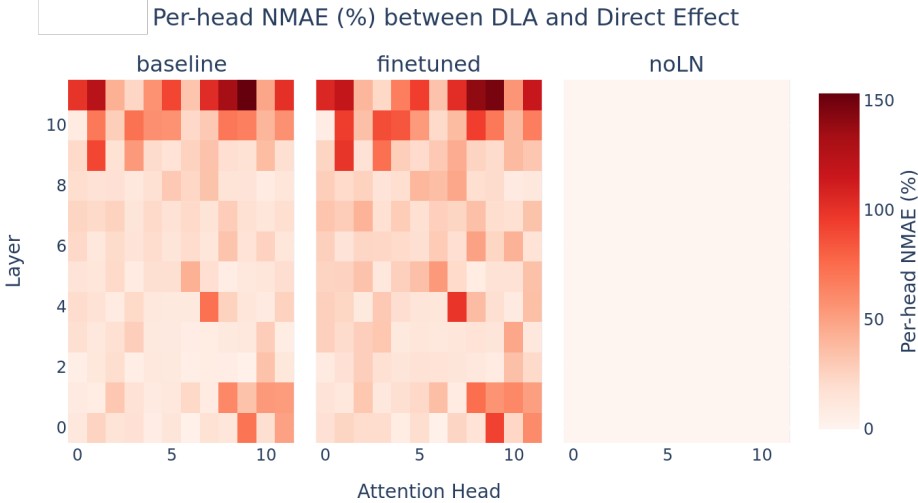

Figure 5: Per-head NMAE (%) between DLA and Direct Effect across all attention heads in GPT-2 Small models: baseline (left), vanilla fine-tuned (middle), and LN-free fine-tuned (right). Significant deviations occur across most attention heads in models with LN. The LN-free model shows no difference across all heads, demonstrating that DLA and DE are equivalent.

## H CONFIDENCE NEURONS

As mentioned in Section 5.4, confidence neurons exhibit two key characteristics: (a) high weight norm, implying importance despite weight decay regularization, and (b) approximately constant contribution to all next token logits, suggesting minimal impact on token prediction. These seemingly contradictory characteristics are reconciled by the final LN, between $\mathbf{w}_{out,i}$ and the unembedding matrix $\mathbf{W}_U$. The effect of confidence neurons on output logits is mediated by this normalization, a mechanism absent in our LN-free models.

These neurons regulate confidence by writing high-norm vectors that project onto an effective nullspace of the unembedding matrix. When these vectors increase the residual stream norm, the final LN scales everything down uniformly, making the output distribution more uniform while preserving token rankings. To identify (b), neurons that preserve token logits ranking, we followed Stolfo et al. (2024) and calculated LogitVar($\mathbf{w}_{\text{out},i}$), the variance in the normalized projection of the neuron's weights with each token in the unembedding matrix:

$$\text{LogitVar}(\mathbf{w}_{\text{out},i}) = \text{Var}\left(\frac{\mathbf{W}_{\text{U}}\mathbf{w}_{\text{out},i}}{\|\mathbf{W}_{\text{U}}\|_{dim=1}\|\mathbf{w}_{\text{out},i}\|}\right). \quad (11)$$

where $\|\mathbf{W}_{\text{U}}\|_{dim=1}$ is the vector of norms of columns of the $W_U$ matrix. Confidence Neurons (CN) maximize the ratio of (a) and (b):

$$\text{CN}(i) = \frac{\|\mathbf{w}_{\text{out},i}\|}{\text{LogitVar}(\mathbf{w}_{\text{out},i})}. \quad (12)$$

Figure 6 summarizes CN identification in both GPT-2 Small and GPT-2 Medium models: the same identical set confidence neurons persist as across all model variants (we chose to highlight the top-7), including LN-free models where their theorized mechanism of action is absent. These neurons maintain their characteristic high weight norm and low logit variance signature despite fine-tuning and even the removal of LN.

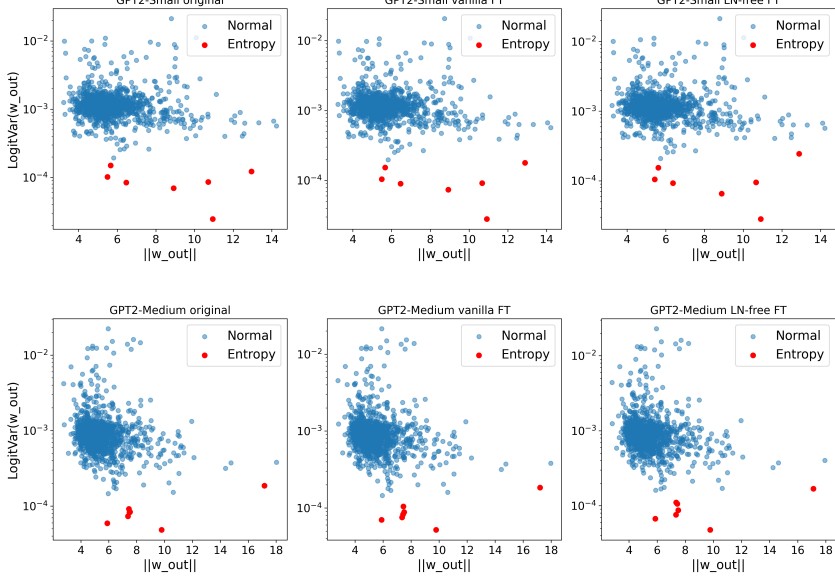

Figure 6: Identification of confidence neurons in GPT-2 Small (top) and GPT-2 Medium (bottom) across different model variants: original pretrained models (left), vanilla fine-tuned models (middle), and LN-free fine-tuned models (right). The same confidence neurons (highlighted in red) persist across all model variants, exhibiting characteristically high weight norms and low logit variance.

Having observed identical confidence neurons across all model variants, we next investigated whether their effective nullspaces were modified by performing Singular Value Decomposition (SVD) on

each model's unembedding matrix. Figure 7 shows the normalized singular values (solid lines), revealing similar nullspace patterns, though fine-tuned variants exhibit a slightly smaller effective nullspace. The cosine similarity between top confidence neurons and singular vectors (dashed lines) demonstrates these neurons predominantly project onto the nullspace in all variants, with some non-negligible overlap in transitional regions where singular values approach zero. This may explain why our vanilla fine-tuned model has less effective confidence regulation when mean ablated. Interestingly, the LN-free model maintains an almost identical nullspace and cosine-similarity pattern to the vanilla fine-tuned model, despite having no ability to affect logit rankings.

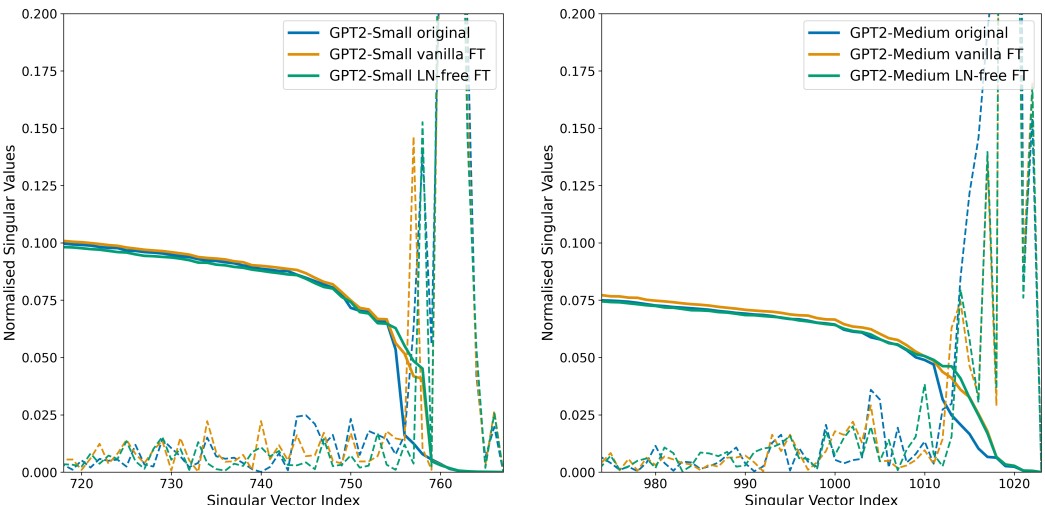

Figure 7: SVD of the unembedding matrix for GPT-2 Small (left) and GPT-2 Medium (right) across model variants. Solid lines show normalized singular values, revealing similar nullspaces across variants, though fine-tuning appears to make the effective nullspace slightly smaller. Dashed lines represent the cosine similarity between the top confidence neuron (584 for Small, 1083 for Medium) and each singular vector. These neurons predominantly interact with the nullspace in all variants, with overlap in regions where singular values approach zero in the fine-tuned models.

To test whether confidence neurons maintain their functional impact across model variants, we performed mean ablation on these neurons (similar to the total effect described in Stolfo et al. (2024)), and measured the resulting change in cross-entropy loss. Figure 8 shows the absolute change in loss when ablating the top-3 confidence neurons in each model. The original GPT-2 Small and GPT-2 Medium models exhibit substantial variation when these neurons are ablated. Without the context-specific LN scaling these neurons provide, the models predicted logit distributions significantly change. The vanilla fine-tuned models show reduced but still notable effects, suggesting these neurons are less effective due to our fine-tuning strategy. This reduced effectiveness may be related to the slightly smaller effective nullspace, though further investigation is needed to confirm this relationship. The LN-free models show almost no variation, implying that these neurons have no effective mechanism to impact final logits despite maintaining their structural characteristics.

To empirically verify that confidence neurons primarily work by modifying the entropy of outputs, we cumulatively ablated the top three confidence neurons in GPT-2 Medium across all variants. Figure 9 illustrates the results. In the original model, ablating all three neurons decreases entropy by over 3% while changing cross-entropy loss by only 0.1%—a 30x difference in magnitude. The vanilla fine-tuned model shows a similar but reduced effect, consistent with our earlier observations of its slightly degraded confidence regulation capability. Again, the LN-free model exhibits no change in either metric. These results directly demonstrate that confidence neurons function by modulating distribution entropy through LN scaling, with minimal impact on which tokens are predicted, allowing them to regulate model uncertainty without changing token rankings. We also investigated whether cumulative confidence neuron ablation of GPT-2 Small vanilla fine-tuned model could yield identical CE loss and entropies to the LN-free model. While the entropies matched (approximately 2.785) when ablating the top-3 neurons, there remained an absolute difference of approximately 0.06 (2%)

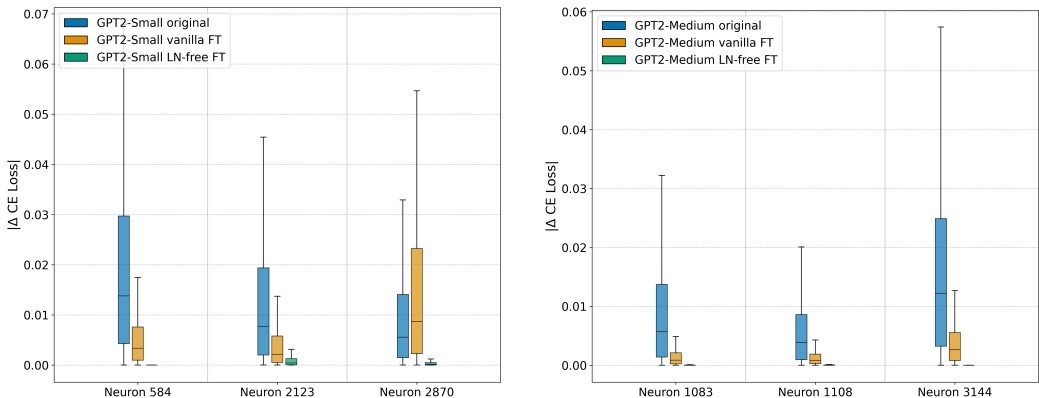

Figure 8: Change in CE loss upon mean ablation of top-3 confidence neurons for GPT-2 Small (left) and GPT-2 Medium (right). The original models (blue) show substantial loss changes when these neurons are ablated, indicating their significant role in confidence regulation. The vanilla fine-tuned models (yellow) exhibit reduced but still notable effects. The LN-free models (green) show almost no change in loss when the same neurons are ablated, confirming that without LN, they lack the mechanism to directly affect output logits.

in CE loss, implying that the general trend of overconfidence in LN-free models arises from more complex mechanisms beyond simply disabled confidence neurons in the final MLP.

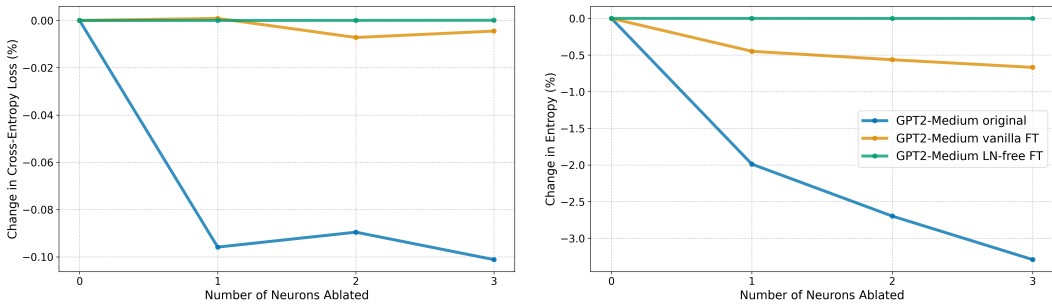

Figure 9: Cumulative effect of ablating the top three confidence neurons in GPT-2 Medium. Left: Relative change in CE loss. Right: Relative change in entropy. The original model (blue) shows a disproportionately large impact on entropy compared to CE loss, demonstrating these neurons primarily regulate distribution confidence rather than token predictions. The vanilla fine-tuned model (yellow) shows reduced effects, while the LN-free model (green) shows no measurable change in either metric.

## I  IMPACT STATEMENT

Our work investigates the role of Layer Norm in transformer-based language models, showing that it can be entirely removed from all GPT-2 models with minimal performance loss. This contributes to the broader interpretability agenda by removing nonlinearity and reducing complexity and entanglement. Our results do not move the frontier of model capabilities; thus, we do not expect our work to create novel risks. In contrast, our work may support safer and more transparent model development by making more tractable and accurate mechanistic interpretability techniques. As with other interpretability advances, there remains the possibility that our work could be used to develop more capable AI systems. However, we believe the release of LN-free GPT-2 models will primarily serve researchers working to understand model internals and improve the transparency of current architectures.

## J STATEMENT ON LLM ASSISTANCE.

We used LLM-based coding assistance to draft tests and make small code edits. While writing the manuscript, we used LLM tools to polish wording, improve clarity, and generate LaTeX math expressions. All suggestions were reviewed and integrated by the authors.

