# OpenReview forum: "Small Transformers Don’t Need LayerNorm at Inference Time: Scaling LayerNorm Removal to GPT-2 XL and Implications for Mechanistic Interpretability"
_ICLR.cc/2026/Conference — ICLR 2026 Poster_

### Official Review · Reviewer_vbJk · 2025-10-20

**Soundness:** 3
**Presentation:** 3
**Contribution:** 3
**Rating:** 6
**Confidence:** 4

**Summary:**

This paper investigates how LN can be removed from LLMs, and develops a step-wise procedure to do so. By removing LNs one by one, while continuing to (pre-)train using CE loss and an additional loss, the authors manage to remove LNs with minimal harm to model performance. They do so for all GPT-2 models and Pythia 70m. The authors then study 2 mech interp techniques thought to be hindered by LN and find that one performs better without LN (DLA) and another does not (AtP). The authors also discover that the confidence neurons discovered by prior work no longer exist in LN-free models.

**Strengths:**

This paper (assuming it is extension of work I have seen previously) tackles an original question: can we remove LN from LLMs? It answers the question in the positive, quite clearly showing that LN can be so removed. The experiments fairly convincingly show that models' overall performance is not greatly harmed, and that this makes certain mech interp techniques more accurate. This is somewhat significant, as LN is generally considered an inconvenience by mech interp researchers.

**Weaknesses:**

- This paper could go a little further in proving that LN-less models are (almost) just as good as ones with LN. For example, it could show that such models can also be fine-tuned or quantized.
- This paper doesn't seem to address fine-tuned / instruction-tuned / RLHF'd / reasoning models at all, even though they are quite often studied by mech interp researchers. Similarly, the models that the paper does address are rather small (though I think that's reasonable, considering the cost of using larger ones). It also doesn't study any models that use RMSNorm.
- In general, my main concern about this paper is its practicality. Even if the number of extra training steps grows sublinearly with model size, this procedure seems inconvenient and compute-intensive to perform. Moreover, as much interpretability work derives its value from studying real-world models that people actually use, the value of studying these LN-free models will likely be limited by the fact that they will see little practical use. This means that this paper will probably be an academic curiosity more than a game-changing development; however, I still think it's interesting enough to publish.

**Questions:**

- Do you have any results for / did you consider the variants of attribution patching (integrated gradients, AtP*)? They seem worth discussing
- Why use IOI for attribution patching? It seems like you could really just use any data (and even better, a more general dataset), since attribution patching can approximate the effect of any activation patching experiment.
- Where exactly do you show that "the amount of fine-tuning data needed for LN removal grows sublinearly with model parameters" (22-23)?
- Comment: Change 318-19 to say something like "Nanda (2023a) describes it for this reason as "a particularly thorny nonlinearity". The current formulation feels a little too much like leaning on Neel Nanda's particular authority as a well-known individual, which is not super appropriate for an academic paper (or otherwise).
- messed-up parentheses in citations (079-80)

---

> ### Author Response · Authors · 2025-11-21
> **Reply to reviewer vbJk (pt1)**
>
> We’d like to sincerely thank you for the thoughtful review and recognition of our work's originality and significance. We address your questions and concerns below.
>
> **Regarding showing LN-free models can be fine-tuned or quantized:**
> As you pointed out, our results show that our LN removal procedure mostly preserves performance. We primarily provide these LN-free models as tools for evaluating interpretability research methods or study LN.
> We don’t want to claim our LN-free model could now be fine-tuned without encountering any stability issues, as we still recognize LN's role in stabilizing training. However, we consider it plausible that fine-tuning LN-free models using an auxiliary loss (similar to what we use during LN removal) could work. In a sense, this is what we have already been doing during our LN removal procedure.
> Regarding quantization, while it should be properly tested empirically, we expect it should be possible. After fine-tuning, activation vectors are not normalized yet our models learn to produce activations with mostly uniform norms across sequence positions (Fig. 3). We don’t expect to find outliers in specific activation space directions of the LN-free models any more than in models with LN. This suggests quantization should work comparably in LN models, although we agree empirical validation is needed to properly confirm this. We are gonna run an experiment to test this.
>
> **Regarding generalization to instruction-tuned or RLHF models:**
> Our analysis focuses on base pretrained models, and we agree that a significant amount of meaningful mechanistic interpretability research is conducted on further finetuned models (instruction-tuned, RLHF'd, reasoning models).
> Generally speaking, we believe our conclusions would generalize to finetuned models (as we find it unlikely for LN to be non-essential for general language modeling but essential for modeling instruction-following or reasoning behavior) and we think of the LN-free models we have open-sourced mostly as testbeds for evaluating mech-interp research techniques.
>
> **Regarding generalization to RMSNorm models:**
> This point was raised also by another reviewer. We believe our procedure will generalize without issue to RMSnorm models due the fact that RMSNorm and LN are almost exactly equivalent.
>
> LN differs from RMSNorm in three ways:
> *  It contains a bias term, which can however be "folded", i.e. reabsorbed in the parameters of an adjacent linear operation.
> * It removes the $(1,1,1,\ldots)$ component of the activation vector before rescaling.
> * Instead of dividing by RMS, it rescales the activation vector by the standard deviation of its components. However, after removing the $(1,1,1,\ldots)$ projection, this is equivalent to rescaling by RMS up to a factor of $\sqrt{d_{\text{model}}}$ that can be folded away.
> Since these normalizations are applied before every model component and before the unembedding, effectively it is as if the LN models do not use the $(1,1,1,\ldots)$ dimension of their residual stream and apply RMS on the remaining dimensions.
>
> Hence we believe our results would generalize to models with RMSNorm instead of LN.
>
> We’ll add this discussion of the two normalization schemes more explicitly in an upcoming revised manuscript version, referencing Gupta [2024] as reference.
>
> Reference:
>
> Gupta, A., Ozdemir, A., & Anumanchipalli, G. (2024). Geometric Interpretation of Layer Normalization and a Comparative Analysis with RMSNorm. arXiv preprint arXiv:2409.12951.

---

> ### Author Response · Authors · 2025-11-21
> **Reply to reviewer vbJk (pt2)**
>
> **Regarding variants of AtP:**
> We did not perform analyses of these variants. Our focus was on standard attribution patching as is the most common and simplest technique and was already discussed in relation to LN’s effect. We agree however that it’d be valuable future work to test these analyses on our LN-free models.
>
> **Regarding why we used IOI for attribution patching:**
> We opted for IOI as it is an established example used for introducing attribution patching [Nanda, 2023a]. This setting allows us to compare results against known ground-truth activation and attribution patching results, and has the additional benefit of helping us to verify that our fine-tuning procedure has not substantially altered the model's underlying circuits. Overall, we believe that our results provide sufficient evidence to support our conclusion, but we also agree that running the experiment on a more general dataset could further corroborate it.
>
> **Regarding where we show sublinear scaling (lines 22-23):**
> Thanks for pointing out this issue. We agree that this claim should be supported more explicitly in the manuscript. The evidence is present in Fig.1/Table3 but not explicitly highlighted. From our results:
>
>
> * GPT-2 Small (124M params): \~157M tokens (\~300 steps × \~524k tokens/step)
> * GPT-2 XL (1.5B params): \~413M tokens (\~800 steps × \~516k tokens/step)
>
> This shows a ~12x increase in parameters required only a ~2.6x increase in fine-tuning tokens. In the upcoming manuscript revision we’ll make this clearer.
>
> **Regarding changes in lines (318-319 and 79-80):**
> Thank you for flagging these issues. We fixed both of them in the revised version of the manuscript we will upload in the upcoming days.
>
> Thank you again for your positive assessment and constructive feedback! We hope that our replies addressed your questions and we will post an updated version of the manuscript in the upcoming days.

---

> > ### Comment · Reviewer_vbJk · 2025-11-24
> >
> > Thanks for this response! I agree that the extensions to RMSNorm/RLHF'd models will probably work (though in the latter case, I wonder what kind of data you'd have to use - would you have to actually use RLHF data, just like you've used LM data here?). I've noted the lack of response to my question about practicality, but I think that this finding is interesting despite that issue. I think this paper would make a good poster.

---

### Official Review · Reviewer_L48A · 2025-10-29

**Soundness:** 3
**Presentation:** 3
**Contribution:** 4
**Rating:** 8
**Confidence:** 4

**Summary:**

This article demonstrates that LayerNorm (LN) is not essential for Transformer inference, introducing a novel fine-tuning protocol that successfully removes all LN layers from the entire GPT-2 family (up to 1.5 billion parameters) with only a minor performance drop. The resulting LN-free models make a significant contribution to mechanistic interpretability, rendering Direct Logit Attribution (DLA) mathematically exact and reducing its approximation error from nearly 50% to 0%. The study also uses these models to causally confirm that "attention sinks" and "confidence neurons" are adaptive mechanisms dependent on LN's non-linearity. Finally, the work reveals LN's previously under-appreciated role in robustness and calibration, as the LN-free models become overconfident.

**Strengths:**

1. Nonlinearities in transformer architectures are a major challenge for mechanistic interpretability. This work takes an important step toward addressing that issue by replacing LayerNorm with a linear alternative.
2. It not only provides a method for removing LN without significantly impacting the model’s overall performance, but also releases a suite of LN-free models that could be valuable to the mechanistic interpretability community.
3. It performs a thorough empirical evaluation to show that model capabilities remain largely intact after removing LN, based on both perplexity and benchmark results.
4. It provides insightful results about the impact of LN on common mechanistic interpretability techniques like direct logit attribution and attribution patching.
5. It also details the impact of the removal of LN on first token norm, attention sink, and confidence neurons.
6. It is well well-written paper which is easy to follow and provides the required details.
7. I also appreciate the broader direction of conducting mechanistic interpretability research to modify and potentially improve model architectures.

**Weaknesses:**

1. As noted in the paper, relying on a single family of models and not evaluating larger models raises concerns about the broader effectiveness of the proposed LN removal method, especially given the fine-tuning involved and the need for extensive hyperparameter tuning. For example, both GPT-2 and Pythia models use standard LayerNorm, whereas many newer models employ RMSNorm, which limits the immediate applicability of the findings to architectures such as Llama or Mistral.
2. While the impact of removing LN is assessed from multiple perspectives. There is more to be done to fully understand what LN actually does and how it helps in language modeling. For instance, this work showed that removal of LN makes the model overconfident, which needs to be better understood and mitigated.

**Questions:**

1. Could you explain how you separated LN_{q,k}​ and LN_{v}​? Since the input to the entire attention block is typically normalized, wouldn’t that mean the keys, queries, and values all receive the same normalized input?
2. Have you tried training a reasonably large language model without LN? If so, how does its performance compare to LN-enabled versions?
3. Beyond the mechanisms discussed in the paper, do you think the removal of LN could significantly affect any known circuits or mechanisms? For instance, could the IOI circuit become more distributed due to the absence of LN?
4. Researchers often apply the final LayerNorm when using LogitLens. Do you think that would no longer be necessary if LN were removed?

---

> ### Author Response · Authors · 2025-11-21
> **Reply to reviewer L48A**
>
> We’d like to thank you for your positive assessment of our and of impact of our contributions. Below we address your questions and concerns:
>
> **Regarding separation of LN_{q,k}​ and LN_{v}​:**
> To do so we simply double the LN layers and then during fine tuning we replace them with the FakeLN linearized versions. After the first replacement of LN with FakeLN queries and keys receive only linearly rescaled activations, whereas values received them normalized (until LN_{v} replacement). LN layers are computationally cheap, so doubling them up poses no compute issues, and once replaced the potentially different rescaling factors can be absorbed in corresponding value and query/key matrices.
>
> **Regarding training LLM without LN from scratch:**
> If we correctly interpreted the question concerns whether we tried training a model (from scratch) instead of only fine tuning it. This point was raised also by another reviewer and we provide a common answer here.
> Training from scratch without LN typically leads to instabilities and doesn't work, however acknowledge that
> Nabeshima [2024] demonstrated this is possible for toy models (and Zhu [2025] from larger ones, but replacing it with a $\tahn$ nonlinearity),
> Despite the fact that LN has an important stabilization role during training, it's possible that a combination of other techniques such as our auxiliary loss and skipping gradients above a certain threshold could enable training larger models without LN.
> We decided to focus on creating fine-tuned LN-free versions of existing models for the following reasons:
> * Training LN-free from scratch would require much more compute than our fine-tuning procedure.
> * Given the training stabilization role of LN, we find it unlikely that this will work even in combination with other techniques, and in any case it would require significantly more effort.
> * Fine-tuning enables better controlled comparisons: as fine-tuning preserves most learned features, providing fine-tuned LN-free versions of existing models enables comparative analyses that better isolate the effects of LN removal.
>
> Finally, we think that training LN-free models from scratch would address an interesting but complementary question (is LN necessary for learning to model language vs. can LN be removed from existing models).
> Regarding mechanism changes due to LN removal:
> Within the paper we show how the presence or absence of LN affects some mechanisms,
> most importantly first token norm and confidence regulation via entropy neurons. However, for IOI, although we have not reconstructed the entire circuit, our partial evidence suggests that the same model components are responsible for implementing the same circuit, in line with the reasoning that by only fine-tuning the model, most circuits are preserved.
>
> **Regarding model overconfidence:**
> While we show how removing LN increases removes confidence regulation via entropy neurons, we don’t believe this is the main reason behind the increase in confidence, which might be connected to other details of our protocol. We discuss this a bit more in the Limitation section 6.1 and in appendix G.
> While we consider an in depth exploration of this phenomenon interesting we also believe it is beyond the scope of this publication.
>
> **Regarding RMSnorm models:**
> We believe our procedure will generalize without issue to RMSnorm models due the fact that RMSNorm and LN are almost exactly equivalent.
>
> LN differs from RMSNorm in three ways:
> * It contains a bias term, which can however be "folded", i.e. reabsorbed in the parameters of an adjacent linear operation.
> * It removes the $(1,1,1,\ldots)$ component of the activation vector before rescaling.
> * Instead of dividing by RMS, it rescales the activation vector by the standard deviation of its components. However, after removing the $(1,1,1,\ldots)$ projection, this is equivalent to rescaling by RMS up to a factor of $\sqrt{d_{\text{model}}}$ that can be folded away.
> Since these normalizations are applied before every model component and before the unembedding, effectively it is as if the LN models do not use the $(1,1,1,\ldots)$ dimension of their residual stream and apply RMS on the remaining dimensions.
> Hence we believe our results would generalize to models with RMSNorm instead of LN.
>
> We’ll add this discussion of the two normalization schemes more explicitly in an upcoming revised manuscript version, referencing Gupta [2024] "Geometric Interpretation of Layer Normalization and a Comparative Analysis with RMSNorm"
>
> **Regarding final LayerNorm application in LogitLens:**
> You are right, it will no longer be necessary, provided that the constant rescaling factor of the final LayerNorm is absorbed in unembedding linear matrix operation (as it is in our models).
>
> Thank you again for your positive assessment! We hope that our replies address your questions!  We will post an updated version of the manuscript in the upcoming days.

---

> > ### Comment · Reviewer_L48A · 2025-11-25
> >
> > Thank you for clarifying my questions. I would like to keep my original score.

---

### Official Review · Reviewer_KPh1 · 2025-10-31

**Soundness:** 3
**Presentation:** 3
**Contribution:** 3
**Rating:** 8
**Confidence:** 3

**Summary:**

The authors show that all layernorm layers in a transformer can be removed via fine-tuning from every GPT-2 model with only a small increase in validation loss (e.g. +0.03 cross-entropy loss for GPT-2 XL). They then study these models in interpretability tasks, showing downstream effects such as improvement in logit-attribution interpretations, improvements to approximation methods like attribution patching, and inactivity of confidence neurons. The work is novel because authors replace LN with a purely linear alternative as compared to earlier works which use non-linear alternatives.

**Strengths:**

I enjoyed reading this paper – I learned something from it. I think others will too!
- The authors carefully dissect LayerNorm layers, find the sequence of removing these layers with several hyper-parameters, and provide a linear transformation equivalent whose loss is only minimally/negligibly worse as compared to the original. They then rigorously test out the effect of removing the layer-norm on important ideas in the literature – such as the effect on direct-logit-attribution as well as attribution patching, attention sinks, confidence neurons. The authors introduce a simple auxiliary loss term to stabilize loss during fine-tuning and do extensive empirical tests to find the best hyper-parameters for the sequence in which Layer-Norms should be removed.
- Authors find that this removal can be scaled sub-linearly to larger models.
- The work suggests exciting new directions of research, on changes and improvements to the transformer architecture with an interpretability lens.

**Weaknesses:**

1. The empirical cost of finding the hyper-parameters for LN removal seems to be really high, and the benefits of this removal don’t seem to be significant. Could training a transformer with the fake-LN be a better approach, rather than figuring out the best way to fine-tune away from it?
2. The results are predominantly on GPT-2 – unclear how this transfers to very large LMs that are in use today.
3. Unclear what the main takeaway is, in terms of whether the LN should be removed or not. The predominant advantage seems to be that logit attribution can be direct, but is the whole process of empirically understanding the best sequence/hyper-parameters to remove the LN worth the trade off?

**Questions:**

1. Did the authors consider training models without LN? How would this compare to their fine-tuned variant as well as the original GPT-2 model?
2. What is the main takeaway for interpretability researchers?

---

> ### Author Response · Authors · 2025-11-21
> **Reply to reviewer KPh1**
>
> We’d like to thank you for your positive assessment and thoughtful feedback. We’re pleased to see that you found the paper enjoyable and recognize its novelty.
> We address your concerns below:
>
> **Regarding empirical cost and training LN-free models:** This question was raised also by another reviewer (we provide a common answer here). Training from scratch without LN typically leads to instabilities and doesn't work, however acknowledge that
> Nabeshima [2024] demonstrated this is possible for toy models (and Zhu [2025] from larger ones, but replacing it with a $\tahn$ nonlinearity),
> Despite the fact that LN has an important stabilization role during training, it's possible that a combination of other techniques such as our auxiliary loss and skipping gradients above a certain threshold could enable training larger models without LN.
> We decided to focus on creating fine-tuned LN-free versions of existing models for the following reasons:
> * Training LN-free from scratch would require much more compute than our fine-tuning procedure.
> * Given the training stabilization role of LN, we find it unlikely that this will work even in combination with other techniques, and in any case it would require significantly more effort.
> * Fine-tuning enables better controlled comparisons: as fine-tuning preserves most learned features, providing fine-tuned LN-free versions of existing models enables comparative analyses that better isolate the effects of LN removal.
>
> Finally, we think that training LN-free models from scratch would address an interesting but complementary question (is LN necessary for learning to model language vs. can LN be removed from existing models).
>
> **Regarding generalization to larger models:**
> We acknowledge this point as a limitation and a future extension of our work. We faced practical constraints in compute and time that limited our ability to explore scaling beyond GPT-2 XL for the time being. However, we have reasons to believe this procedure would scale to larger models:
> * Our current protocol holds across architectures spanning two orders of magnitude in parameters.
> * Although we observed an increase in instabilities in larger models, our auxiliary loss was effective in mitigating them (and could prove effective also in even larger ones, although we acknowledge some uncertainty).
> * The number of fine-tuning steps increases sublinearly with model parameters.
> These points support the feasibility of scaling the method to larger models.
> Nonetheless, we also recognize that scaling to larger models may come with additional challenges (as was scaling from small to XL) and might require further refinements of our protocol (such as optimizing LN removal gaps or exploring alternative versions of the auxiliary loss), which we identify as promising future work in Section 6.2.
>
> **Regarding main takeaway:**
> Our primary claim is that normalizing nonlinearities such as LN are not necessary for transformers to effectively model language at inference. By demonstrating that LN can be removed post-training with minimal performance loss, we show that LN's role is primarily in training stabilization rather than being fundamental to language modeling capabilities.
>
> We document our protocol and open source LN-free models as a tool for interpretability research to enable more precise mechanistic analyses when LN's nonlinearity becomes particularly problematic. One such use could be in cleaner decomposition into sparse circuits, as component contributions are now truly disentangled. Another interesting insight our work revealed is the following: contrary to field expectations, LN is not a primary factor hindering attribution patching accuracy.
>
> More broadly, we think that comparing models with and without LN could shed light on mechanisms implemented by LN and on their importance (or lack of necessity).
>
> **Regarding the empirical cost:**
> While we recognise that hyperparameter tuning required effort, we openly released both the protocol and trained models to allow the community to use LN-free GPT-2 variants without repeating this process and/or to explore further protocol improvements.
>
> Thank you again for your positive review.

---

> > ### Comment · Reviewer_KPh1 · 2025-11-21
> >
> > Noted, thanks for your responses, and congratulations on producing this fun paper :)

---

### Official Review · Reviewer_6KTD · 2025-11-01

**Soundness:** 4
**Presentation:** 4
**Contribution:** 3
**Rating:** 8
**Confidence:** 4

**Summary:**

The authors show that LayerNorm modules can be removed from GPT-2 models through fine-tuning, with only a small increase in validation loss across all model sizes. The resulting LN-free models are released on Hugging Face. They also conduct a series of experiments to examine how removing LayerNorm affects several well-known phenomena (such as direct logit attribution, attribution patching, first-token high L2 norm, and confidence neurons). Their analysis reveals that some of these phenomena originate from the presence of LayerNorm, while others remain unaffected by its removal.

**Strengths:**

* The paper is very well written and easy to follow.
* The experiments and baselines are thoughtfully designed and make it easy to understand the impact of each decision involved in fine-tuning the LayerNorm-free models. The approach is evaluated across multiple GPT-2 model sizes, strengthening the conclusions.
* The paper analyzes several interpretability techniques and phenomena to examine the effects of removing LayerNorm. This enables a controlled investigation into whether well-known behaviors in language models are driven by LayerNorm itself or emerge from other underlying mechanisms. The analyses are well executed, and the conclusions are very interesting.
* all their LN-free models are open-sourced and publicly available on Hugging Face

**Weaknesses:**

* A key limitation is that the analysis remains largely focused on the GPT-2 family. Although the authors include results for Pythia-70M, this provides only limited evidence of generality, and it is unclear whether the findings extend to newer or larger architectures.
* The largest model evaluated is GPT-2 XL (1.5 B parameters), which is relatively small.
* The title (“Transformers Don’t Need LayerNorm at Inference Time”) feels overclaiming, as the experiments are limited to GPT-2 models and up to 1.5B parameters.
* Some mathematical notation is introduced without adequate definitions.

**Questions:**

I think the paper is very interesting and well written.
* Did you attempt to train an LN-free model larger than GPT-2 XL? It would be helpful to understand whether the proposed method scales beyond 1.5B.

---

> ### Author Response · Authors · 2025-11-21
> **Reply to reviewer 6KTD**
>
> We'd like to thank you for your thoughtful review. We really appreciate your positive assessment of our paper's clarity, experimental design, and the interesting nature of our findings. We address your concerns below:
>
> **Regarding scope and generalization of results:**
> We acknowledge this is a limitation of our results and an opportunity for interesting future work, as we point out in Section 6.2. We faced practical constraints in compute and time that limited our ability to train models beyond GPT-2 XL (adapting our pipeline across model families requires more effort than scaling within a family).
> However, we believe our findings provide meaningful evidence for generalization:
> * Across architectures: We focused primarily on GPT-2 models due to their ubiquity in the interpretability community. We then tested the method on Pythia-70M, which, while small, has substantially different architectural choices from GPT-2 (RoPE, parallel attention and feed-forward layers, different initialization schemes). Our method generalized out of the box on this different architecture.
> * Across scales: Our results hold consistently across GPT-2 Small, Medium, Large, and XL, spanning two orders of magnitude in parameters (117M to 1.5B), with sublinear scaling in finetuning data with model size. As we discuss in Section 6.1, we observed that instabilities become more frequent in larger models, but regularization methods such as our auxiliary loss we employed proved effective in mitigating them at the scale we tested. We acknowledge some uncertainty here, but we believe these results support scaling feasibility.
>
> **Regarding title:**
> We understand this concern. The title choice was meant to capture our central finding: that LayerNorm is not fundamentally required at inference time to model language effectively with transformers. We chose to emphasize this conceptual insight while also explicitly acknowledging the scope of our validation ("Scaling LayerNorm Removal to GPT-2 XL"). We care about transparency regarding the scope of our results, which is why we specified both the method and the model scale in the title. However, we are open to revising it if these considerations are insufficient.
>
>
> **Regarding mathematical notation:**
> To make the mathematical content clearer, we implemented the following changes in the upcoming version of the manuscript:
> * Explicitly mentioned that $\odot$ in the LN formulas represents an elementwise operation, in this case over the residual stream dimension.
> * Renamed $H$ to $N_{\text{head}}$ in the NMAE formula to avoid nomenclature duplicates with $H$ as residual stream model dimension.
> * Clarified that $\|W_U\|_{\text{dim}=1}$ in the LogitVar formula represents a vector of norms computed along the columns of the $W_U$ matrix.
>
> Could you please point us to any other specific notation that needs clearer definition? We want to ensure that all mathematical content is accessible in the manuscript revision.
>
> Thank you again for the constructive feedback.

---

> > ### Comment · Reviewer_6KTD · 2025-11-26
> >
> > Thank you for the detailed response! I’ll keep my original score.

---

### Author Response · Authors · 2025-12-02
**Authors final remark to AC**

In this work we show that LayerNorm, long considered a hindrance for mechanistic interpretability of LLM, can be gradually removed through finetuning with minimal performance loss. We apply common mechanistic interpretability techniques to study how the presence or absence of LayerNorm impacts them, and open-sourced LayerNorm-free GPT2 models to enable the mech-interp community to conduct cleaner circuit analysis and investigate LayerNorm’s role.

We were pleased to see that all reviewers evaluated our work positively (scores 8, 8, 8, and 6), praising its presentation (“paper is very well written and easy to follow”), its experimental design (“the experiments and baselines are thoughtfully designed”), and its contribution to the field (“the conclusions are very interesting”). Reviewers also appreciated the open-sourcing of our LN-free models. All reviewers rated soundness, presentation, and contribution as good or excellent.

Alongside the positive evaluations, reviewers raised some concerns, which we addressed in our replies and by expanding the following points in the revised manuscript (changes highlighted in blue):

* We expanded the Conclusions section to better reflect the scope and the practical implications of our work.
* We added an appendix to more clearly highlight the data supporting the sublinear scaling behaviour of required fine-tuning data with model size.
* We added an appendix section to discuss the relationship between LayerNorm and RMSNorm, and our expectations on generalization to RMSNorm models.
* We added a note in the Limitations sections  that the models we open-sourced are not easily quantizable via HF transformers (which is in any case uncommon in mech-interp research).
* We added explanations regarding mathematical notations to enhance clarity.

We thank the reviewers for their constructive feedback and believe the revisions implemented strengthen the paper’s clarity and more accurately represent the scope and practical implications of our contributions.

---

### Meta-Review · Area_Chair_cHS3 · 2026-01-19

**Summary:**

Given the uniformly positive reviews, this decision should have been straightforward. Instead, it required unusually long deliberation. Consequently, this is likely the most “rejection-like” meta-review I have ever written for a paper I am ultimately recommending for acceptance.

My hesitation is driven almost entirely by the paper's severe over-claiming and the misleading nature of its main title. “Transformers Don’t Need LayerNorm at Inference Time” makes an extraordinarily broad and categorical claim. What the paper actually demonstrates is far narrower: some small transformer models can tolerate replacing LayerNorm with linear substitutes at inference time, provided they are carefully fine-tuned. As multiple reviewers have noted, all evaluated models are substantially smaller than the transformers commonly used today. There is therefore no evidence that the conclusions generalize to modern large-scale transformers, or even to transformers in general. Presenting such limited empirical findings under an absolute and universal claim is misleading.

Even within the studied regime, the claim “don’t need” is itself inaccurate. The reported performance differences are marginal, and model stability is sensitive to fine-tuning choices. This is not a case where LayerNorm is shown to be unnecessary, but rather one where its role can sometimes be partially approximated under restrictive conditions. The distinction between “can be pruned with careful fine-tuning” and “do not need” is substantial, and the paper consistently adopts the latter framing. This choice of tone is not a minor stylistic issue; it contributes to a broader and damaging culture of exaggerated claims in the field, and risks misleading readers who do not closely inspect the experimental scope.

That said, the paper’s empirical contribution is genuinely strong. The study is systematic, carefully executed, and reflects substantial experimental effort. While I have reservations about comparing architectures using identical training steps and learning-rate schedules despite their differing computational characteristics, the evidence is nevertheless sufficient to establish a concrete and interesting point: LayerNorm can, in some settings, be pruned or approximated without catastrophic degradation. Although the proposed method is currently computationally expensive, the work has real potential to stimulate follow-up research on normalization pruning and inference-time simplification.

In summary, I am recommending acceptance despite the paper’s framing, not because of it. I strongly believe that the title and the tone of the main claims must be substantially softened to accurately reflect the scope and limitations of the results. Without such changes, the paper’s presentation risks causing more harm than its technical contribution justifies.

**Reviewer Concerns:**

* not comparing using bigger models or other models: not fully resolved except for studying Pythia, but understandable (with the paper's subtitle)
* overclaim of title: not resolved, as discussed above
* training without LN instead of finetuning: not resolved
* interpretability takeaway and more insights: reasonably explained. There is always room for improvement but the current contents look sufficient.
* mathematical notation and other minor issues: resolved mostly

**Reviewer Scores:**

Most of the "bigger" issues are clear limitations fo the work and were not resolved. So I don't think the scores will be increased. But they were already high in the beginning.

---

### Decision · Program_Chairs · 2026-01-26

Accept (Poster)